# hTERT-Driven Immortalization of RDEB Fibroblast and Keratinocyte Cell Lines Followed by Cre-Mediated Transgene Elimination

**DOI:** 10.3390/ijms22083809

**Published:** 2021-04-07

**Authors:** Nadezhda A. Evtushenko, Arkadii K. Beilin, Erdem B. Dashinimaev, Rustam H. Ziganshin, Anastasiya V. Kosykh, Maxim M. Perfilov, Alexandra L. Rippa, Elena V. Alpeeva, Andrey V. Vasiliev, Ekaterina A. Vorotelyak, Nadya G. Gurskaya

**Affiliations:** 1Center for Precision Genome Editing and Genetic Technologies for Biomedicine, Pirogov Russian National Research Medical University, Ostrovityanova 1, 117997 Moscow, Russia; hopeevt@gmail.com (N.A.E.); arkadii.beilin@gmail.com (A.K.B.); dashinimaev@gmail.com (E.B.D.); avkosyh@gmail.com (A.V.K.); 2Koltzov Institute of Developmental Biology of Russian Academy of Sciences, 26 Vavilova Str., 119334 Moscow, Russia; rippa86@yandex.ru (A.L.R.); alpeeva_l@mail.ru (E.V.A.); 113162@bk.ru (A.V.V.); vorotelyak@yandex.ru (E.A.V.); 3Shemyakin-Ovchinnikov Institute of Bioorganic Chemistry, Miklukho-Maklaya 16/10, 117997 Moscow, Russia; rustam.ziganshin@gmail.com (R.H.Z.); sufrep@gmail.com (M.M.P.)

**Keywords:** RDEB, COL7A1, hTERT, bmi-1, fibroblast, keratinocyte, immortalization, 3D skin equivalent, stratification, secretomes, Cre recombination

## Abstract

The recessive form of dystrophic epidermolysis bullosa (RDEB) is a crippling disease caused by impairments in the junctions of the dermis and the basement membrane of the epidermis. Using ectopic expression of hTERT/hTERT + BMI-1 in primary cells, we developed expansible cultures of RDEB fibroblasts and keratinocytes. We showed that they display the properties of their founders, including morphology, contraction ability and expression of the respective specific markers including reduced secretion of type VII collagen (C7). The immortalized keratinocytes retained normal stratification in 3D skin equivalents. The comparison of secreted protein patterns from immortalized RDEB and healthy keratinocytes revealed the differences in the contents of the extracellular matrix that were earlier observed specifically for RDEB. We demonstrated the possibility to reverse the genotype of immortalized cells to the state closer to the progenitors by the Cre-dependent hTERT switch off. Increased β-galactosidase activity and reduced proliferation of fibroblasts were shown after splitting out of transgenes. We anticipate our cell lines to be tractable models for studying RDEB from the level of single-cell changes to the evaluation of 3D skin equivalents. Our approach permits the creation of standardized and expandable models of RDEB that can be compared with the models based on primary cell cultures.

## 1. Introduction

Recessive dystrophic epidermolysis bullosa (RDEB) is a rare genetic disease mainly characterized by extreme skin fragility caused by mutations in the *COL7A1* gene encoding type VII collagen (C7). Disruption in the function of this protein leads to separation of sublamina densa induced by minor trauma that causes blistering and erosion of the skin, itching and pain. Persistent inflammation accompanying these lesions results in development of skin squamous carcinoma in 30–50-year-old RDEB patients [1].

Unfortunately, in addition to its rarity, RDEB is also heterogeneous. Currently, there are more than 800 known different mutations in the *COL7A1* gene, with a wide profile of their localization and types of mutations affecting severity of cutaneous manifestation [2,3]. These problems hurdle the development of effective treatments. Nevertheless, new directions in drug development and gene therapy of RDEB constantly appear and require rigorous testing. At the same time, the availability of patient biomaterials is severely limited, both due to the small number of patients and their health condition. This makes the development of RDEB models an important task required to accelerate the testing of new approaches to treatment. The drawbacks of primary cell lines as a valuable model system are their heterogeneity and patient-specific genetic differences, restricted availability and limited lifespan.

Skin is the main source of primary cultures in this case. Immortalization of fibroblasts is relatively widely spread and easier. The ectopic expression of the catalytic subunit of the human telomerase gene (*hTERT*) is sufficient to significantly extend the lifespan of human fibroblasts. The transfer of exogenous *hTERT* cDNA in cells circumvents telomere-controlled senescence without inducing global chromosomal instability [4,5,6]. Several studies showed that restoring telomerase activity is not sufficient for the immortalization of human primary cells of certain types without acquiring the additional karyotype aberration or tumor transformation [7]. So, the properly immortalized keratinocyte lines are few. The most widely used are HaCaT, Normal Immortal KeratinocyteS (NIKS) and N/TERT 1, N/TERT 2G; all of them were obtained due to spontaneous immortalization in laboratory conditions and possessed a karyotype aberration [8,9,10,11]. However, directed and controlled immortalization of keratinocytes is still challenging. Immortalized cell cultures could be established by heterological expression of viral oncogenes, such as large T antigen of the simian virus 40 (SV40) virus or the E6/E7 antigen of the human papilloma virus (HPV); however, in many cases, this type of transformation causes the appearance of cancer-associated traits [12,13,14]. Immortalization with the HPV E6 and E7 genes could lead to obtaining cells with properties close to the primary ones but differing in several aspects, including downregulation of plasma membrane-associated proteins [15]. However, this approach was applied to establish a tractable model of RDEB HPV E6/E7-transformed keratinocyte lines [16]. Here, we propose lentivirus-mediated hTERT/BMI-1 immortalization to obtain new immortalized patient-specific cell lines, which retain their cell type-specific properties.

The basic cascades of cell senescence and, consequently, the target pathways for immortalization, are p16INK4A/pRb and p14ARF/p53. Often, p53 and pRb are activated by products of the INK4a/ARF locus encoding two tumor suppressors: p16(CDKN2a/INK4a) and p14(ARF). P16 encodes the inhibitor of kinases that regulates the progression of cells through the G1 phase; it is expressed in normal non-transformed cells and becomes inactivated during neoplastic transformation [17]. Polycomb complex protein BMI-1 represses expression of the INK4A/ARF locus; downregulation of the *bmi-1* gene induces both physiological and premature aging in keratinocytes, whereas *bmi-1* overexpression inhibited p16INK4A activation and increased the clonogenic capacity [18]. This property allowed the use of BMI-1 for immortalization of epithelial cells [19,20].

In previous work, we characterized several new primary RDEB fibroblast lines [21]. The orphan state of RDEB and its heterogeneity contribute to the uniqueness of each of the cases described. This study is focused on the transformation of these primary cell lines into the stable ones preserving the features of RDEB models. The establishment of new immortalized cell lines, fibroblasts and keratinocytes capable of retaining normal growth and differential characteristics in vitro extends the instrumental tools in experimental dermatology research.

## 2. Results

### 2.1. Description of the Experimental Material

In the present work, we used primary cell lines (Table 1) obtained in our previous work and added new ones, Keratinocytes of Epidermolysis Bullosa patient (KEB) [21]. Among the four RDEB patients, d1 and d3 possessed the manifestations of a generalized severe RDEB form, d2 had a RDEB inversa (RDEB-I) form and d4 featured RDEB of moderate clinical severity.

### 2.2. Immortalization of Fibroblast and Keratinocyte Cell Lines

The immortalization of FEB and FHC lines was made by stable expression of *hTERT* cDNA. After lentiviral transduction, the cells were cultivated in the presence of puromycin and, after 7–10 passages, analyzed for the presence of integrated constructs.

#### 2.2.1. Genetic Constructs

A lentiviral plasmid encoding hTERT TurboFP635NLS was constructed (Figure 1A). Furthermore, the auxiliary lentiviral plasmid encoding the Azurite fluorescent protein was constructed in the same way, except for using the Azurite cDNA insert instead of the *hTERT* cDNA in the first plasmid. The presence of a pair of LoxP (locus of X-over P1) repeats in the lenti plasmids makes it possible to apply Cre-mediated site-directed recombination to cut out the floxed insert, i.e., *hTERT* or *Azurite* cDNA (Figure 1B). KEB and KHC were immortalized by both hTERT and BMI-1 overexpression. Didier Trono provided the pLOX-CWBmi1 plasmid (Addgene plasmid #12240). Lentiviral transduction was followed by cell selection by their growth with the presence of puromycin. 

#### 2.2.2. Immunocytochemistry (ICC) Analysis of hTERT and BMI-1

Immortalized cultures were tested for hTERT and BMI-1 expression by ICC. The results showed positive staining against hTERT in immFEB and immFHC lines and against hTERT and BMI-1 in immKEB and immKHC lines (Figure 1C).

#### 2.2.3. Telomerase Activity in Immortalized Cell Lines

TRAPeze. The activity of the ectopically expressed telomerase in the new panel of immortalized cell lines was tested by the TRAPeze assay. This assay used the “TRAP” (Telomeric Repeat Amplification Protocol) principle which is based on the two subsequent reactions catalyzed by telomerase and Taq polymerase respectively. The number of fragments and their respective lengths depend on the number of discrete TRAP steps which, in turn, correspond to the telomeric repeats. Thus, the activity of telomerase in cell extracts is proportional to the number of fragments revealed by TRAP [22]. It should be mentioned that the number of TRAPeze products should not be considered as the number of telomeric repeats of the cells, although they have a direct proportional correlation. The results of TRAPeze quantitative polymerase chain reaction (qPCR) with the TSR8 (quantitation control template) quantification standard provide the dynamic range of the measurement of telomerase activity in cell extracts, which is defined as an amol value. The near linear range of the log plot with dilutions of TSR8 from 0.0016 to 0.2 amol and cell extracts containing equal amounts of total protein was considered. No telomerase activity was detected in the corresponding primary cell lines while the activity of telomerase in newly obtained cell lines was comparable to the common immortal cell lines: 3T3 (NIH), HELA and HaCaT (Figure 1D,E), therefore confirming functional activity of the inserted genetic construct.

Capillary electrophoresis of TRAPeze probes. The lengths of TRAPeze qPCR fragments were estimated using capillary gel electrophoresis. The accurate resolution of qPCR fragments allows detecting the difference between immortalized and primary cells in terms of the number of amplified fragments as well as the length of the last detectable fragment. Appendix A shows the distribution of amplified fragments for one of the pairs of FEB lines. We detected a much greater length of TRAPeze qPCR fragments in all immortalized cell lines compared to their primary progenitors (Appendix A).

#### 2.2.4. Western Blot Analysis

Western blot (WB) analysis of hTERT. The expression of hTERT in immortalized dermal fibroblasts was analyzed by WB (Figure 1F). We detected intense staining of the hTERT band in all immortalized cell lines comparable to HaCaT and almost no staining in primary cells.

WB analysis of BMI-I. The expression of BMI-I in immortalized keratinocytes was analyzed by WB (Figure 1G). We detected intense staining of the BMI-I band in all immortalized cell lines and almost no staining in primary cells.

### 2.3. Morphology and Functional Analysis of Immortalized Fibroblasts

#### 2.3.1. General Morphology

We investigated the impact of immortalization on morphology of fibroblasts. From 43 to 127 cells from each cell line were measured for morphological analysis by phase contrast microscopic imaging and more than 10 thousand cells—by flow cytometry. Morphology and scatter parameters of primary and immortalized cells from the same donors were compared using the two-sample *t*-test and one-way ANOVA with the Tukey’s test, respectively (Figure 2).

Next, we tested whether immortalized RDEB fibroblasts remain different from immortalized healthy fibroblasts in some parameters as it was shown previously for primary fibroblasts [21]. We compared combined data from four immFEB to immFHC by the two-sample *t*-test for microscopy data and by the FlowJo Chi Squared comparison for flow cytometry data (Figure 2).

In spite of some cell lines showing differences in some morphological parameters, there was no clear dependence in any parameter between morphology and immortalization. For most parameters, we can say that immortalization does not have a significant impact on fibroblast morphology therefore maintaining differences between FEB and FHC such as an increased area, perimeter, fit ellipse minor axis, minimum caliper diameter, size and granularity for FEB (Figure 2).

#### 2.3.2. Fibroblast Marker Expression

We characterized FEB and immFEB lines by ICC analysis for fibroblast marker expression (Figure 3) and did not obtain any evidence regarding any difference in their expression including diminished expression of C7.

#### 2.3.3. Collagen Contraction by Immortalized Fibroblasts

The contraction of four collagen gels was measured for each immFEB line (1, 2, 3 and 4) and the respective lines of primary state. Comparison by the Mann–Whitney test did not show significant differences in fibroblast-induced collagen contraction between primary and immortalized cells (Figure 4A).

#### 2.3.4. Senescence-Associated-β-Galactosidase Staining in Fibroblasts

Primary and immortalized fibroblast pairs were stained with X-gal to compare their senescence-associated-β-galactosidase (SA-β-gal) expression. The results show that after immortalization, fibroblast staining intensity became closer to primary fibroblasts from a young donor and, in most cases, significantly lower than in the paired primary cells (Figure 4B).

### 2.4. Cre-Mediated Splitting Out of Transgenes

#### 2.4.1. Transient Transfection of the Target and Auxiliary Cell Lines

The presence of a pair of LoxP repeats in lenti plasmids makes it possible to apply Cre-mediated site-directed recombination to cut out the floxed insert, i.e., *bmi-1*, *hTERT* (or *Azurite* cDNA in the model experiment).

The Cre-LoxP system was used by transient transfection of the cells of interest and fluorescence imaging of them the next few days after transfection. In this case, the transfected cells acquired a red fluorescence signal (TurboFP635 NLS) in the nucleus after Cre-mediated recombination. To check the differences of TurboFP635NLS expression in cells before and after Cre-mediated recombination, we used transient transfection of these cells by the pCre-eGFP plasmid. The population of cells with red fluorescent nuclei appeared 24 h after transfection and became dominant after a few days (Figure 5(A1,A2)). Fluorescence-activated cell sorting (FACS) analysis of immFEB cells before and after Cre-eGFP transfection is shown in Figure 5B. Next, qPCR analysis of genomic DNA samples revealed a decrease in the abundance of the hTERT gene in cells after Cre-eGFP transfection. For this, DNA from immFEB1 cells with and without Cre transfection was analyzed for the relative amount of the hTERT–TurboFP635 junction and TurboFP635 amplicons. The relative quantities of hTERT–TurboFP635 junction amplicons to standalone TurboFP635 amplicons of immFEB1 before and after Cre transfection were calculated by the ΔΔCt method (Figure 5C). 

The effect of Cre-mediated recombination was demonstrated for Human Embryonic Kidney 293T cell line (HEK293T) with the stable expression of the auxiliary construct, Azurite LoxP TurboFP635 NLS. After transduction, the cells acquired blue fluorescence; the stable line HEK293T Azurite LoxP TurboFP635 NLS was selected and demonstrated bright blue fluorescence evenly distributed in cells. Transient transfection of these cells by the Cre encoding plasmid leads to obtaining the population of cells with red fluorescent nuclei (Figure 6(A1,A2)). FACS analysis of HEK293T Azurite LoxP TurboFP635 NLS before and after Cre recombination is shown in Figure 6B. Amplification of DNA fragments flanking TurboFP635 in cells after Cre transfection confirmed transgene splitting (Appendix A). 

#### 2.4.2. Cellular Senescence after Splitting Out of Transgenes

We checked the senescence of cells after deletion of the hTERT cassette, comparing them also with primary cells. In order to obtain a more reliable and accurate evaluation of the senescent state, we combined the staining for senescence-associated beta galactosidase (SA-β-gal) with Ki-67 ICC. The experiment was carried out on the immFEB1 and immFEB4 cells before and after transfection by the Cre recombinase encoding plasmid. We showed the increased intensity of SA-β-gal staining in Cre-transfected cells, which acquired a visual evidence of hTERT knockout—the appearance of red fluorescent nuclei after transfection (Figure 7A).

The percentage of cells with red fluorescent nuclei was counted after several rounds of Cre transfection and amounted to 41% for immFEB1 and 13% for immFEB4. The proliferation of immFEB with red fluorescent nuclei was estimated based on ICC staining against Ki-67. The immFEB1 and immFEB4 with red fluorescent nuclei showed decreased proliferation compared to FEB and immFEB without Cre transfection (Figure 7B). The number of passages was 15 for the FEB line and 30 for immFEB, both with and without Cre transfection.

### 2.5. Morphology and Functional Analysis of Immortalized Keratinocytes

#### 2.5.1. General Morphology

We investigated the impact of immortalization on morphology of the keratinocytes KEB1, KEB2 and KHC from the donors d1, d2 and d12, respectively. From 193 to 455 cells from each cell line were measured for morphological analysis. Morphological parameters of primary and immortalized cells from the same donors were compared by the two-sample *t*-test. FSC and SSC flow cytometry data were compared by the FlowJo Chi Squared comparison. At the moment of the experiment, immortalized cells had undergone 12–18 passages more than primary keratinocytes.

No statistically significant differences in any morphological parameters were found between keratinocyte lines when analyzing morphology in a 2D culture in phase contrast images. Flow cytometry data revealed that in all three cases, immortalized cells were slightly smaller by FSC (T(x) = 651–1253, baseline T(x) = 10–46) and less granular by SSC (T(x) = 463–1518, baseline T(x) = 4–29) than their primary progenitors. Furthermore, there were two populations of cells by SSC in immortalized cultures and a single population in primary cultures (Appendix A).

Comparison of RDEB keratinocytes to the healthy control revealed no difference both for primary and immortalized cells (T(x) < 354).

#### 2.5.2. Stratification of Immortalized Keratinocytes In Vitro

We reconstructed epidermis in vitro using our immKEB2 and immKHC8. Fully stratified epidermis was obtained for each of the cases. Basal and suprabasal keratinocytes express cytokeratin 14. Granular layer keratinocytes express loricrin and are located near desquamating cells. Plectin expression is located mainly near the basal zone. Staining against C7 reveals differences in expression level (Figure 8). ImmKEB2 line originated from keratinocytes of the donor’s skin with the RDEB Inversa form of disease, for whom the residual amount of C7 protein was shown [21].

#### 2.5.3. Tumorigenicity of Immortalized Keratinocytes In Vivo

Tumorigenicity of the immortalized keratinocytes was tested in the xenograft animal model. The cells were transplanted into immune-deficient Nude (NU/NU) mice into testes as immune privileged organs to minimize the immune response [23].

All three mice from the control group with HELA cell transplants formed tumors within 3–4 weeks after operation. Mice with tumors were sacrificed after tumor size prevented the animal from having a free range of motion. The size of the testes reached 3.2 cm in diameter, primary tissue analysis showed the presence of cysts with liquid. Further, we confirmed the presence of morphological changes in the testis tissues using hematoxylin–eosin staining (Figure 9A). ImmunoHistoChemistry (IHC) staining for human nuclei proved that HELA cells formed tumors (Figure 9B).

Testes with immKEB2 cell transplants were collected 10 weeks after operation. Examination showed that the testes were of normal size; tissue analysis also did not reveal structural changes in the organs (Figure 9C). A small number of living cells of the graft were found with human nuclei staining (Figure 9D).

Based on the above, we confirmed that the immKEB2 cells, despite the fact that they survive in 10-week-old transplants in immunodeficient mice, are non-tumorigenic.

#### 2.5.4. Western Blot Analysis of Type VII Collagen in Keratinocytes

We performed Western blot analysis of C7 expression for keratinocytes. The results showed an about 10-fold difference in C7 staining intensity between RDEB and healthy keratinocytes (Figure 10A).

#### 2.5.5. Keratinocyte Proliferation

Immortalized keratinocytes from all three lines were shown to have increased proliferation when compared to their primary states that we showed by 5-Ethynyl-2-Deoxyuridine (EdU) labeling and flow cytometry. It was shown that the proliferation rate for immortalized cells is 20%, 16% and 13% higher for immKEB1, immKEB2 and immKHC8, respectively, compared to their primary cells (Figure 10B).

#### 2.5.6. SA-β-gal Staining in Keratinocytes

We performed SA-β-gal staining of primary and immortalized keratinocytes. More than four thousand cells were analyzed for each cell line. ImmKEB1, immKHC8 and immKEB2 were shown to have increased, decreased or no changed in staining intensity compared to the corresponding primary cells (Figure 10C). Therefore, no changes in SA-β-gal activity correlating with the juvenile state of immortalized cells were found. Furthermore, no correlation between SA-β-gal activity and RDEB state of the keratinocytes was found.

#### 2.5.7. Secretome Analysis of immKEB1, immKEB2 and immKHC

Data processing of the lists of differentially expressed proteins (DEPs) was carried out with software packages of MaxQuant (Appendix A). The complete list of differential extracellular proteins consisted of 134 and 157 significantly up- and downregulated proteins in RDEB keratinocytes, respectively (Pp0.05, Benjamini–Hochberg-corrected). The filtration of data was made using the base quantitative analysis of the empty medium, then only the proteins with log_2_
*t*-test difference of more than 1 (>1) and less than −1 (<−1) were selected.

When the initial list of secreted proteins was narrowed down to only those that changed significantly, the question of their functional relevance arose, which we addressed through bioinformatic resources. Three lists of DEPs were obtained, for pairs immKEB1 and immKHC (55 proteins), immKEB2 and immKHC (22 proteins) and immKEB1 and immKEB2 (9 proteins). There were common members of DEPs in each of the groups, several proteins were specific only for one of the groups. We analyzed the pooled list of DEPs for each immKEB relative to immKHC (Figure 11A (left), immKEB1 + immKEB2 versus immKHC) and a list of DEPs specific to both immKEB lines (Figure 11A (right), immKEB1 ∩ immKEB2 versus immKHC). Then, we analyzed proteins that were differentially expressed in either KEB1 or KEB2 lines (Figure 11A (left), immKEB1 + immKEB2 versus immKHC) in comparison with KHC and proteins, which were specific for both RDEB lines (Figure 11A (right), immKEB1 ∩ immKEB2 versus immKHC). These two lists of proteins were denoted as 1 + 2_3 for immKEB1 + immKEB2 versus immKHC and 1 ∩ 2_3 for immKEB1 ∩ immKEB2 versus immKHC. Principal component analysis (PCA) was applied to assess the quality of the dataset. The samples of the same origin (biological repeats) were clustered together on the PCA plot (Figure 11B). Two groups of proteins (1 + 2_3 and 1 ∩ 2_3) were analyzed further by software Search Tool for the Retrieval of Interacting Genes/Proteins (STRING) [24] and PANTHASUS [25].

The protein interaction network made by STRING is shown in the Appendix A. Heatmaps, principal component analysis and gene ontology (GO) analysis with EnrichR were completed in Phantasus v1.9.2 (Figure 11A,B, Appendix A). Expression of proteases and inhibitors of proteases, proteoglycans and matrisome-associated glycoproteins were revealed, GO terms revealed groups specific for adhesion and regulation of peptidase activity, regulation of response to wounding, remodeling of the extracellular matrix.

## 3. Discussion

To create a model system for studying RDEB, a panel of cell lines with extended lifetime was developed by immortalization via exogenous *hTERT* cDNA or a combination of it with *bmi1* cDNAs. The important properties of the model system were to retain the most initial traits of the primary cells. The expansible cultures of fibroblasts and keratinocytes displayed the properties of their progenitor cells. The majority of primary cells entered the senescence after several dozens of divisions in culture. Meanwhile, the cells being differentiated could not maintain the ability to proliferate. In this respect, the establishment of cell populations with the capacity to proliferate but preserving the metabolic activity and the initial signatures is quite an urgent problem [26].

Forced expression of hTERT was applied widely to establish immortalized cell lines, particularly to expand the lifetime of dermal fibroblasts [27]. However, recent data have revealed that hTERT-elongated telomeres could lead to dramatic chromosome mislocalization [28]. Taking this into account, we performed short tandem repeat (STR) profiling for all of new RDEB immortalized lines (Appendix A). This analysis showed that immFEB1 and immFEB2 have normal distribution of STR markers coincidental with those for immKEB1 and immKEB2 lines. No chromosomal aberration was found in three immFEB lines that were investigated for it.

The success of hTERT insertion and its expression was supported by the TRAPeze assay, which showed a significant increase of telomerase activity (Figure 1D,E) and extended length of the TRAPeze qPCR fragments (Appendix A) in immFEB and immKEB. WB and ICC also clearly showed the presence of the telomerase protein in cell lysates (Figure 1F) of immFEB as well as of BMI-1 in immKEB (Figure 1G). 

We had shown previously that primary FEB lines differ from FHC in C7 expression, ability to contract collagen gel and several morphological parameters [21]. In this work, we showed that immFEB does not differ in these properties from primary FEB which they originate from (Figure 2, Figure 3 and Figure 4A).

Keratinocytes in a 2D culture show no differences in morphology after immortalization but flow cytometry showed the appearance of small and less granular cells in cultures of immKEB and immKHC (Appendix A). Furthermore, the increased proliferation rate was shown by EdU labeling (Figure 9D). Since the enhanced expression of hTERT is observed in most cancers and the expression of BMI-1 has been reported in various types of cancer, including squamous cell carcinoma [29], we decided to test one of the newly established cell lines for the ability to induce tumors in an immunodeficient mouse model, but no tumor formation was observed (Figure 9A).

The main property of keratinocytes besides specific marker expression is the ability to form stratified epidermis. The comprehensive investigation of skin cell properties is impossible without establishing the 3D modeling, namely, 3D skin equivalents. The development of skin organotypic structure began in 1981, when Bell et al. [30] described a tissue-engineered 3D human skin equivalent consisting of dermal and epidermal layers. A variety of strategies have been developed so far in the area of skin bioengineering. Here, we established skin equivalents from immKEB and immKHC by the method we used for primary keratinocytes [31]. The results obtained indicate that our immortalized keratinocytes fully retained the ability to undergo successive stages of differentiation and stratification with onset of appropriate marker expression. Components of cornified cell envelopes were shown in suprabasal layers proving high approximation of differentiation capabilities of immKEB to those of immKHC (Figure 8).

Fibroblasts and keratinocytes both produce C7; the latter’s secretion accounts for a more significant share of the Anchoring Fibril (AF) component. It was shown that C7 expression in keratinocytes is higher than in fibroblasts [32]. Suggesting that fibroblast secretion of C7 is not abandoned while the difference in the amount of protein might not be clearly visible, the plausible difference in expression was checked by analysis of C7 in keratinocytes. The expression of C7 was shown for the immKHC skin equivalent, while the major downregulation in C7 expression was revealed for immKEB2 skin equivalents demonstrating about half of staining intensity compared to a healthy skin equivalent (Figure 8). The KEB2 cells belonged to the d2 patient with the RDEB-I form of disease, for whom the residual expression of COL7A1 was shown earlier [21]. While IHC staining of C7 was not that specific, the results of WB (Figure 10A) highlighted the differences in expression between immKEB and immKHC lines, semi-quantitatively estimated as a circa 10-fold decrease of the C7 content in KEB.

Generally, the rejuvenation of immortalized cells was shown by analyzing β-galactosidase activity which was believed to be associated with senescence [33]. The SA-β-Gal assay in the panel of lines FEB–immFEB confirmed the more juvenile state of almost all immFEB lines with one exception (Figure 4B). In the case of immFEB1, the intensity of staining increased. The reason for that could be a difference in the passage number. ImmFEB1 had about 30 passages more than the primary FEB1 cells, whereas for other pairs, this difference was about 10 passages. Still, the immFEB1 staining was comparable to early passage FHC4 (seven passages) (Figure 4B). 

It was reported that SA-β-gal staining has its own limits connected with false positive cases due to the serum starvation of cells, composition of the medium or more complex processes involving lysosomal activity. In contrast to immFEB, we did not show any difference between immortalized keratinocytes and their progenitor primary cells in SA-β-gal staining (Figure 10C). It was shown previously that this SA-β-Gal assay can be combined with other biomarkers, as SA-β-gal was also expressed when cells were maintained at confluence, when both quiescent and senescent cells were SA-β-gal-positive [34,35]. The activity of SA-β-gal is independent of DNA synthesis. Proliferation rate is another factor correlated with senescence of cell cultures [34,36]. Staining fibroblasts against Ki-67 revealed no correlation between the percentage of Ki-67-positive cells in primary and immortalized FEB1 lines, but demonstrated difference in case of FEB4 (Figure 7B), whereas labeling of keratinocytes by EdU revealed a clear increase in the percentage of dividing cells in immortalized keratinocytes lines (Figure 10B).

Upon insertion of *hTERT* cDNA, the target construct consisted of two directly orientated LoxP sites, flanking a telomerase cassette located downstream of a human phosphoglycerate kinase 1 (pGK) promoter and upstream of red fluorescent protein (TurboFP635NLS) coding regions (Figure 1A,B). The insertion of *hTERT* surrounded by LoxP sites makes it possible to split out the insert of this gene from cells in response to Cre activity [37].

A widespread approach of reversible immortalization is the tetracycline-inducible expression of the immortalizing agent in cell lines [38], although the method used in this study has several advantages. The background activity of the tetracycline-dependent promoter has dramatic consequences in the case of the *hTERT* transgene, as a small amount of enzyme is enough to sufficiently elongate telomeres, which imposes restraints on the determination of the immortalization effect and the possibility of clinical development of the method in the first place. Secondly, the additional load of the drug removes the initial phenotype of cells. Using the Cre-LoxP system enables switching the cells’ mitotic potential by exposing it once.

To test the efficiency of this process, we constructed a AzuriteloxPTurboFP635NLS plasmid, made lentiviral transduction and generated a stable HEK293T Azurite cell line. Transient transfection of Azurite-positive cells by the Cre-eGFP plasmid led to accumulation of red fluorescent signals in the nuclei of Cre-positive cells (Figure 6). The switching of the fluorescence signal from blue cytoplasmic to red nuclear occurred due to the cutout of the Azurite insert together with the translation termination codon before the TurboFP635 NLS open reading frame. This experiment showed the potential ability to get rid of the *hTERT* insertion from the new immortalized FEB, KEB and FHC, KHC lines.

To increase the effect, several rounds of Cre transfection could be applied. We subjected immFEB1 to three rounds and immFEB4 to one round of Cre transfections. It resulted in 40% and 13% of cells demonstrating TurboFP635 fluorescence in nuclei in immFEB1 and immFEB4, correspondingly. We performed several tests to check whether splitting out the exogenous cassettes would result in restoration of some of the traits of the primary cells. We showed that after Cre transfection, immFEB cells demonstrate diminished proliferative capacity (Figure 7B) and increased SA-β-gal activity (Figure 7A), therefore confirming at least partial reversal closer to the primary state.

Both the enhanced expression and activity of telomerases are characteristic traits of cancer cells, metastatic and cancer stem cells. Moreover, it is supposed to be connected or partly responsible for the drug resistance phenomenon of cancer cells. 

The RDEB patient-specific cell lines with expansible lifespan, both in culture and in 3D models, represent convenient models for further assessment of the disease mechanism and testing the effects of gene therapeutic approaches [39]. The immortalized panels of FEB and FHC lines provide an unlimited source of materials to investigate the properties of RDEB cells and may highlight new potential targets for RDEB therapeutic approaches.

Two RDEB-specific keratinocytes lines were derived from patients d1 and d2 with generalized severe RDEB and RDEB inversa forms, respectively [21]. Particularly, KEB1 and KEB2, correspondingly, carried a homozygous c.425A < G mutation in *COL7A1* (NC1 domain of C7) and a compound heterozygous mutation of COL7A1 (NC1 and Triple Helical Collagenous (THC) domains of C7). The pattern of DEPs of these two KEB lines was shown to be different (Figure 11A).

The list of DEPs of immKEB2 (versus immKHC) was significantly shorter than that of immKEB1. ImmKEB2 was also closer to immKHC by the first principal component on the PCA plot of the expression data (Figure 11B, upper plot). 

A number of inflammation-related factors were dysregulated for either immKEB1 or immKEB2 lines. Alterations included a decrease in semaphorin 7A (SEMA7A), C–X–C motif chemokine ligand 1 (CXCL1) expression together with an increase in interleukin 1 receptor, type II (IL1R2), receptor 4 for prostaglandin E2 (PTGER4) expression for immKEB1 and an increase in elafin (ELAF/PI3) expression in immKEB2. It was well-known that ELAF is absent in primary adult keratinocytes, highly upregulated in psoriasis skin and was also reported for immortalized N/TERT1 keratinocytes [11].

The matrix turnover system was substantially disrupted according to the expression of proteases, their inhibitors and structural ExtraCellular Matrix (ECM) proteins. We observed the increase of protease inhibitors SERPINB7 (serpin family B member 7) (for immKEB1), SERPINB5 (serpin family B member 5) (for immKEB2) together with the downregulated expression of HtrA serine peptidase 1 (HTRA1). Concurrently, there was an increase in matrix metallopeptidase 1 (MMP1), carboxypeptidase A4 (CPA4) and a decrease in tissue metallopeptidase inhibitor 2 (TIMP2), tissue factor pathway inhibitor 2 (TFPI2) expression for both immKEB lines. Furthermore, lysosomal exoprotease cathepsins were deregulated both in immKEB1 (cathepsin D (CTSD), cathepsin C (CTSC) upregulation) and immKEB2 (cathepsin L2 (CTSL2) downregulation). This profile of expression reflects the special state of RDEB cells with the dysregulated balance of protease expression.

Among structural proteins of basement membranes critical for the dermal–epidermal junction, collagen type VI alpha 1 chain (COL6A1) for immKEB1 and collagen type XVIII alpha 1 chain (COL18A1) for immKEB2 were downregulated, possibly indicating enhanced protease secretion. Only immKEB2 had a reduced expression of C7-binding partner laminin subunit gamma-2 (LAMC2) as well as the increased expression of its receptor integrin subunit alpha 3 (ITGA3). It was shown earlier that changes in the expression level of ITGA3 and laminin 332 deposition influence migration of immortalized and primary keratinocytes [40]. On the other hand, immKEB1 shows decreased expression of nidogen (NID1), another component of the basal membrane, and enhanced expression of filamin A (FLNA) that mediated the interaction between integrins and actin.

Several changes observed in diseased cells indicate altered migration ability of epidermal cells. We found downregulation of expression of malignancy-related factor chondroitin sulfate proteoglycan 4 CSPG4 (NG2) [41]. This proteoglycan is expressed by dermal papilla tip keratinocytes and promotes keratinocyte motility [42] as well as positively regulates CCN2 [43]. An alteration in the migration process could also be seen in the significant decline in the signal regulatory protein alpha (SIRP-α1, SHPS-1) expression in immKEB lines. The role of this factor is not completely clear for epithelial cells, but there is evidence pointing to the contribution of SIRP-α1 in adhesion and migration signaling pathways [44,45]. Together, the downregulation in these three factors may indicate a decrease in migratory ability due to integrin pathway disruption through the loss of C7. Midkine (MDK) and the neuronal cell adhesion molecule (NRCAM) have a significant decrease in expression, both factors being positive regulators of the epithelial–mesenchymal transition [46,47] which has a link to cell mobility activation and oncotransformation. Nucleobindin 2 (NUCB2) is another decreased factor that is associated with cancer and affects keratinocyte migration in wound healing by its fragment [48,49]. The decrease of these tumor-supporting markers in both immKEB lines could be viewed as anticancer traits, preserving keratinocytes from the early development of SCC.

Another side of the epithelial–mesenchymal transition and signaling related to keratinocyte migration is wound healing and fibrosis. Fibrotic pathways are also affected in RDEB. It was shown previously that RDEB keratinocytes negatively regulated expression of transforming growth factor beta (TGF-β) produced by RDEB fibroblasts [16]. Here, we found that the CD109 molecule has enhanced expression in RDEB immKEB1 and partly in KEB2 lines and hypothesized that it could negatively modulate the TGF-β signaling in keratinocytes. This is supported by the decrease in cellular communication network factor 2 (CCN2, CTGF) in both immKEB lines, as it is a TGF-β-induced fibrosis marker [50,51]. 

Therefore, it was demonstrated that the lack of C7 expression in immKEB lines affected and changed the composition of the ECM and the profile of secreted proteins. Overall, the pattern of observed changes was similar for two lines, but each immKEB line had several specific traits. It is still unclear whether these discrepancies came from the different COL7A1 mutations or resulted from the overall RDEB variability.

## 4. Materials and Methods

### 4.1. Lentiviral Plasmids

For immortalization of primary dermal fibroblasts and keratinocytes, we employed a lentiviral transduction by one or both vectors encoding *hTERT* cDNA and *bmi-1* cDNA. 

The *hTERT* cDNA sequence was amplified from Addgene plasmid #69809 (Research Resource Identifiers (RRID): Addgene_69809, xlox(GFP)hTERT was a gift from David Ott; http://n2t.net/addgene:69809 (accessed on 15 February 2021)) with primers specific to *hTERT* cDNA.

The insert encoding *hTERT* cDNA was cloned into modified plasmid pLCMV-Puro that was previously described in [52]. In this plasmid, the CMV promoter was changed to a relatively less stronger pGK promoter; under its control, a cassette with a pair of LoxP sites surrounded *hTERT* cDNA with a “stop” codon placed upstream of the TurboFP635 NLS (i.e., LoxP *hTERT* “stop” codon LoxP TurboFP635 NLS). TurboFP635 NLS was amplified from the plasmid described in [53]. Purification of plasmid endonuclease digestions and PCR products was made by agarose gel electrophoresis and subsequent extraction with CleanUp Standard Kit (Evrogen, Moscow, Russia). The plasmid for immortalization is referred to as “pGK hTERT loxPTurboFP635NLS”. The second lentiviral plasmid (pGKloxPAzuriteTurboFP635 NLS) was constructed in the similar way except for using the *Azurite* insert instead of *hTERT* which was amplified from Addgene plasmid #36086 (RRID: Addgene_36086, pLV-Azurite was a gift from Pantelis Tsoulfas; http://n2t.net/addgene:36086 (accessed on 15 February 2021)). The plasmid for BMI-1 expression is Addgene plasmid #12240 (RRID: Addgene_12240, pLOX-CWBmi1 was a gift from Didier Trono; http://n2t.net/addgene:12240 (accessed on 15 February 2021)).

### 4.2. List of Specific Primers

The primers specific for the pGK promoter, hTERT and TurboFP635 were used to test for stable integration of the *hTERT* transgene.

Age1hTERT—GCCAACCGGTGCCACCATGCCGCGCGCTCCCGC; EcoR1hTERT—GCCTGAATTCTTAGTCCAGGATGGTCTTGAAGTCTG; pGK—ACCGACCTCTCTCCCCAGGGTCT; hTERT3000—ACAAGATCCTCCTGCTGCAG; hTERT_dir—CCACCAAGCATTCCTGCTCAAGCTGAC; TurboFP635rv—GGATCTGTATGTGGTCTTGAGGGAGC; Turborv—TCAGCTGTGCCCCAGTTTGCTAGGCAGG; Turbo-rev—GTCCAGGATGGTCTTGAAGTCTGAGGGCAGTGCCGG; TurboTYRS-dir—CTCAAGACCACATACAGATCCAAGAA; 

pGKdir5—CCGGACGTGACAAACGGAAG; 

pGK dir 4—CCCTCGTTGACCGAATCACC; 

TurboFP635rev2—CCA GTT TGC TAG GCA GGT CG; 

TurboFP635 rev end—CTCGAGATCCGAGTCCGGATTCATCC.

### 4.3. Cell Cultures

The culture media for fibroblasts was based on DMEM High Glucose (4.5 g/L) (Capricorn Scientific, Ebsdorfergrund, Germany) with 10% fetal bovine serum (Capricorn Scientific, Ebsdorfergrund, Germany), GlutaMAX (Gibco, Grand Island, New York, NY, USA), sodium pyruvate (Gibco, Grand Island, New York, NY, USA) and PenStrep (Gibco, Grand Island, New York, NY, USA). The culture medium for keratinocytes was the CnT-07 medium (CELLnTEC, Bern, Switzerland). Both cultures were maintained in a 5% CO_2_ incubator at 37 °C. Subculture of fibroblasts and keratinocytes was performed using the Versene solution (PanEco, Moscow, Russia) and 0.05% trypsin–EDTA (Gibco, Grand Island, New York, NY, USA). For storage in liquid nitrogen, the cells were frozen in the culture medium with 10% DMSO (PanEco, Moscow, Russia). Before placing in liquid nitrogen, cryovials with cells were cooled down from room temperature to −70 °C at the rate of −1 °C/min.

HEK293T was used for lentiviral production (pLOX-CWBmi1, AzuriteloxPTurboFP635NLS, hTERTloxPTurboFP635NLS lentiviruses) and for generation of stable cell line AzuriteloxPTurboFP635NLS.

### 4.4. Stable Cell Line Development

One day prior to transfection, 4 × 10^6^ HEK293T cells were seeded into 60-mm cell culture dishes (Corning, Corning, New York, NY, USA) in 3 mL of DMEM (Capricorn Scientific, Ebsdorfergrund, Germany) with 10% of fetal bovine serum (Capricorn Scientific, Ebsdorfergrund, Germany). Half an hour before transfection, the medium was changed to 1.3 mL of Opti-MEM (Gibco, Grand Island, New York, NY, USA). Opti-MEM (300 μL) was mixed with 20 μL Lipofectamine 2000 (#11668030, ThermoFisher Scientific, Waltham, MA, USA) and incubated for 5 min. Packaging plasmids pR8.91, 2.0 μg, and pMD.G, 0.6 μg (plasmids were a gift from Didier Trono, http://tronolab.epfl.ch (accessed on 15 February 2021)), together with the transfer plasmid (encoding *hTERT*, *bmi-1* or *Azurite* cDNA), 2 µg, were added to 300 μL of Opti-MEM. The plasmid solution was slowly added to the Lipofectamine 2000–Opti-MEM mixture and mixed. After 15 min of incubation, the mixture was added to HEK293T dropwise. After 4 h of incubation, the medium was changed back to DMEM with 5% of inactivated fetal bovine serum. The virus-containing medium was collected after 48 h, filtered through a 0.45-µm filter and immediately used to infect fibroblasts. After 4 h, the medium of fibroblasts was changed to AmnioMax-II (Gibco, Grand Island, New York, NY, USA) and after one day of incubation, puromycin selection started. The medium of keratinocytes was CnT-07 (CELLnTEC, Bern, Switzerland). Fibroblasts and keratinocytes were grown in the presence of puromycin with a gradually increasing concentration, from 0.2 μg/mL to 0.8–1.0 μg/mL. The cells transduced with pGKloxPAzuriteTurboFP635NLS were tested for the fluorescence signal 3–5 days after cultivation. After 10 days of cultivation, the cells expressing Azurite FP were sorted using FACS (FACSAria cell sorter equipped with a 70 μm nozzle, AmCyan channel for Azurite FP).

### 4.5. Detection of Telomerase Activity

Telomerase activity in immortalized fibroblasts and keratinocytes was detected using a PCR-based TRAPEZE kit (S7710, Chemicon, Temecula, CA, USA) according to the manufacturer’s instructions. Briefly, cell extracts were resuspended in 200 µL of the 3-cholamidopropyl dimethylammonio 1-propanesulfonate (CHAPS) lysis buffer per 1 × 10^6^ cells and incubated for 30 min on ice. The samples were centrifuged at 12,000 rcf for 10 min at 4 °C. The supernatants were transferred to new tubes. Protein concentration was determined by measuring absorbance at 280 nm (NanoPhotometer P360 Implen, Munchen, Germany). The samples were diluted to achieve a protein concentration of 500 ng/μL. qPCR reaction was performed with two units of Hot-Start Taq Polymerase (Evrogen, Moscow, Russia) following the manufacturer’s protocol. Fluorometric detection of telomerase activity was carried out on a CFX96 Real Time System (Bio-Rad, Hercules, CA, USA) with software CFX Maestro (Bio-Rad, Hercules, CA, USA).

Telomerase activity was calculated as the amount of extended telomerase substrate (amoles) produced per mg of protein per minute for each sample cell extract.

The length of qPCR fragments was estimated by capillary gel electrophoresis with a 3500 Genetic Analyzer (ThermoFisher Scientific, Waltham, MA, USA). Each sample was diluted 50–100 times and then precipitated with ethanol in the presence of ammonium acetate; then, the pellet was dissolved in the solution of deionized formamide (Sigma-Aldrich, St. Louis, MI, USA) containing a SY650-labeled ladder (CK0701 Syntol, Moscow, Russia; GeneScan™ LIZ™ dye Size Standard, ThermoFisher Scientific, Waltham, MA, USA).

### 4.6. Western Blot Analysis

The samples were prepared by lysed cell pellets in the RIPA buffer (for hTERT analysis) or a RIPA with 8 M urea (for C7 analysis), diluted in the Laemmli buffer (Bio-Rad, Hercules, CA, USA) 1:1 and separated in an SDS-PAGE gel: 8% gel with 8 M urea (C7), 10% gel (hTERT, BMI-1). Then, the separated proteins were transferred to nitrocellulose (C7) or a 0.45 μm PVDF (hTERT, β-actin, BMI-1) membrane. C7 WB analysis was carried out as described previously [21]. For the analysis of hTERT, BMI-1 and ꞵ-actin semi-dry transfer (15 V, 45 min) in a Tris–glycine buffer (Tris base, 47.9 mM, glycine, 38.6 mM, 10% ethanol) was carried out. The membranes were blocked with 3% (*w*/*v*) nonfat dry milk in Tris Buffered Saline, with Tween (TBST) for 1 h, incubated overnight at +4 °C with a corresponding antibody solution: anti-C7, -hTERT, -β-actin. Then, the membranes were washed, incubated with a corresponding HRP-conjugated secondary antibody and imaged using an ECL kit and an imaging system. The detailed protocols of lysate preparation, Polyacrylamide Gel Electrophoresis (PAAG) and transfer procedures see in [21].

### 4.7. Immunocytochemical Fluorescence Staining

The cells were fixed using 10% buffered formaldehyde (Biovitrum, Saint Petersburg, Russia). The fixed cells were incubated with primary antibodies in a block solution based on Phosphate Buffered Saline (PBS) (PanEco, Moscow, Russia) with 10% fetal bovine serum (Capricorn Scientific, Ebsdorfergrund, Germany) and 0.3% TRITON X-100 (Sigma-Aldrich, St. Louis, Missouri, MO, USA) at 4 °C overnight and then with secondary antibodies in PBS with 0.3% TRITON X-100 for 2 h at room temperature. Nuclei were stained with DAPI (Biotium, Fremont, CA, USA).

The antibodies used were as follows:

Primary anti-collagen I antibodies (RAH C11, Imtek, Moscow, Russia)

Primary anti-collagen VI antibodies (ab6586, Abcam, Cambridge, UK)

Primary anti-fibronectin antibodies (ab2413, Abcam, Cambridge, UK)

Primary anti-S100A4 antibodies (ab27957, Abcam, Cambridge, UK)

Primary anti-collagen VII antibodies (C6805, Merck, Kenilworth, NJ, USA)

Primary anti-Ki-67 antibodies (ab16667, Abcam, Cambridge, UK)

Primary anti-hTERT antibodies (ab230527, Abcam, Cambridge, UK)

Primary anti-BMI-1 antibodies (ab14389, Abcam, Cambridge, UK)

Secondary anti-mouse Alexa-594 (A21201, Invitrogen, Carlsbad, CA, USA)

Secondary anti-rabbit Alexa-594 (A21442, Invitrogen, Carlsbad, CA, USA)

### 4.8. Immunofluorescent Collagen Type VII Expression Assay

Fibroblasts were seeded into the wells of a 96-well plate (Corning, Corning, New York, NY, USA) in amounts of six thousand cells per well. The cells were cultured for seven days and then fixed with 10% buffered formaldehyde and stained with primary anti-collagen VII antibodies (ab93350, Abcam, Cambridge, UK) and secondary anti-rabbit antibodies conjugated with Alexa-594 (A21442, Invitrogen, Carlsbad, CA, USA). Nuclei were stained with DAPI (Biotium, Fremont, CA, USA). Fluorescent microphotographs of the cells were made with an EVOS FL AUTO microscope with the same channel settings for all images.

Images were processed using the FiJi software [54]. Nuclei and background were removed, leaving only the stained cytoplasm of the cells. Staining intensity of the cytoplasm was evaluated as the mean pixel intensity value for each individual image.

### 4.9. Skin Equivalents In Vitro

Skin equivalents were manufactured based on the CELLnTEC protocol [55] with slight modifications. Briefly, fibroblasts were seeded inside 24-well plate inserts (35024, SPL, Pocheon-si, Korea) in a fibroblast culture medium and incubated in a 5% CO_2_ incubator at 37 °C for 7–10 days. Keratinocytes were then seeded onto fibroblasts in a keratinocyte culture medium and incubated for 3–4 days in the CnT-07 culture medium (CELLnTEC, Bern, Switzerland). The culture medium was changed to CnT-Airlift (CELLnTEC, Bern, Switzerland) and the cells were incubated for 3–4 days. The culture medium was removed from the inner part of the inserts and skin equivalents were exposed to the air–liquid interface. The culture medium was changed every 2–3 days for 14–21 days.

### 4.10. Tumorigenicity Operation for Keratinocyte Transplants

Immortalized keratinocytes were transplanted into the testes of NU/NU nude mice (Charles River) according to a protocol previously described in [23]. The animals were kept under Specific Pathogen Free (SPF) conditions with a 12-h light/12-h dark cycle, 19–25 °C temperature range and 40–60% humidity. The experiment was approved by the Local Ethics Committee for Clinical Research of the Pirogov Medical University (No. 16/2019). The animals were anesthetized with 1% isoflurane. Three mice were injected with a 1 × 106 immortalized keratinocyte suspension in 20 μL PBS per mouse. The control group (three mice) was injected with a 1 × 106 HELA cells suspension per mouse. Testicles were collected for IHC analysis after tumor formation in case of the control group or 10 weeks after. 

### 4.11. Frozen Sections and Immunohistochemical Fluorescence Staining

Skin equivalents or testes were fixed in 10% buffered formaldehyde (Biovitrum, Saint Petersburg, Russia) overnight, washed in PBS (PanEco, Moscow, Russia) three times for 1 h and placed in 30% sucrose (Sigma-Aldrich, St. Louis, MO, USA) on PBS overnight. The fixed tissues were embedded in Tissue-Tek O.C.T. (Sakura, Osaka, Japan) and were frozen in liquid nitrogen vapor. Tissue sections were made by criotomy. Slides with mounted sections were air-dried and then stored at −70 °C.

The mounted sections were incubated with primary antibodies in a block solution based on PBS (PanEco, Moscow, Russia) with 10% fetal bovine serum (Capricorn Scientific, Ebsdorfergrund, Germany) and 0.3% TRITON X-100 (Sigma-Aldrich, St. Louis, MO, USA) at 4 °C overnight and then with secondary antibodies in PBS with 0.3% TRITON X-100 for 2 h at room temperature. Nuclei were stained with DAPI (Biotium, Fremont, CA, USA). Glycerol (50%) in PBS was used as the mounting medium.

The antibodies used were as follows:

Primary anti-collagen VII antibodies (ab693350, Abcam, Cambridge, UK)

Primary anti-cytokeratin 14 antibodies (ab181595, Abcam, Cambridge, UK)

Primary anti-loricrin antibodies (ab24722, Abcam, Cambridge, UK)

Primary anti-plectin antibodies (ab32528, Abcam, Cambridge, UK)

Conjugate anti-nuclei, clone 235-1, Alexa Fluor 488 antibodies (MAB1281A4, Sigma-Aldrich, St. Louis, MO, USA)

Secondary anti-mouse Alexa-594 (A21201, Invitrogen, Carlsbad, CA, USA)

Secondary anti-rabbit Alexa-594 (A21442, Invitrogen, Carlsbad, CA, USA)

Secondary anti-mouse Alexa-488 (A11029, Invitrogen, Carlsbad, CA, USA)

Secondary anti-rabbit Alexa-488 (A32790, Invitrogen, Carlsbad, CA, USA)

Secondary anti-chicken Alexa-488 (A11039, Invitrogen, Carlsbad, CA, USA)

### 4.12. Confocal Imaging 

Confocal images were made using an LSM 880 confocal scanning microscope (Carl Zeiss Microscopy GmbH, Jena, Germany) based on the Axio Observer.Z1 Zeiss inverted fluorescent microscope equipped with six laser lines (633, 594, 561, 543, 514, 488 and 405 nm), five objectives (EC Plan-Neofluar 5×/0.16, EC Plan-Neofluar 10×/0.3, positive lock (PL) apochromatic (APO) 20×/0.8, PL APO 40×/0,95, PL APO 63×/1,4 Oil Differential Interference Contrast (DIC)) and Laser Scanning System (LSM) software ZEN 2.

The emission bands used as follows: DAPI, 410–579 nm, eGFP, 493–579 nm, TurboFP635, 582–754 nm, Azurite, 410–504 nm, Alexa Fluor-594, 585–733 nm.

### 4.13. Cell Morphology Analysis

Fibroblasts were seeded in low density so the single cells could be separated from each other on microscopic images. Phase contrast photographs of the cells were made using EVOS FL AUTO the day after seeding. The Phase Contrast Microscopy Segmentation Toolbox (PHANTAST) FiJi plugin [56] was used to segment the cells on the images. Holes in binary masks were filled and shapes were analyzed with a standard FiJi function “Analyse particles.” The following parameters were analyzed: area, perimeter, bounding rectangle (width, height and aspect ratio), fit ellipse (major and minor axes), circularity, roundness, solidity and caliper diameter (max and min).

Keratinocytes were grown till near 100% confluence. Phase contrast photographs of the cells were made using an EVOS FL AUTO microscope. Image processing was performed using MorphoLibJ [57]. Centers of the cells were manually selected using a paint brush tool to create binary marker mask images. Marker-controlled Watershed plugin [58] was used to outline the cell’s borders. The obtained images were analyzed with a standart FiJi function “Analyse particles” excluding objects on the edges. Incorrectly selected regions were manually removed from the samples. The following parameters were analyzed: area, perimeter, bounding rectangle (width, height and aspect ratio), fit ellipse (major and minor axes), circularity, roundness, solidity and caliper diameter (max and min).

Estimation of cell size and granularity was made by analyzing FSC and SSC from flow cytometry data from the Bio-Rad S3 Cell Sorter (Bio-Rad, Hercules, CA, USA).

### 4.14. Collagen Gel Contraction

Fibroblasts were embedded into a gel made out of rat tail collagen type I [59] with the collagen concentration of 3 mg/mL and the fibroblast concentration of 1 × 10^5^ cells/mL. Liquid gel (500 μL) with cells was placed into each well of 24-well plates. After gelation, 500 μL of the culture media were added into each well. The next day, the gels were detached from the walls of the wells using syringe needles and allowed to contract. All of them were scanned using an EVOS FL AUTO microscope on day 2 and the area of each collagen tablet was measured using the FiJi software [54].

### 4.15. Cre-Mediated hTERT/Azurite Cutout

Addgene plasmid #13776 (RRID: Addgene_13776, pCAG-Cre:GFP was a gift from Connie Cepko; http://n2t.net/addgene:13776 (accessed on 15 February 2021)) was used for transient transfection of immortalized fibroblasts. After transfection, the cells were grown for two days and the appearance of cells with red fluorescent nuclei was visualized by fluorescence microscopy using an EVOS FL AUTO microscope. The selection of TurboFP635 NLS-positive cells was made by FACS. Twenty-four hours before transient transfection of fibroblasts or HEK293T by the Cre-GFP plasmid, the cells were seeded on glass bottom culture dishes (Fluorodish, World Precision Instruments, Sarasota, FL, USA). For transfection, Lipofectamine 2000 (Promega, Madison, WI, USA) was used in accordance with the manufacturer’s protocol. The transfection mixture was diluted with 700 μL of OptiMEM (Gibco, Grand Island, New York, NY, USA), and incubation was continued for 4–6 h.

For imaging HEK293T, Azurite LoxPTurboFP635 transfected by pCAG-Cre:GFP Leica DMI6000b (Leica Microsystem, Heidelberg, Germany) was used. This inverted microscope is equipped with an HC PL APO 40 × 0.85 lens and HCX PL FLUOTAR L 20 × 0.40 objective finder (Leica Microsystem, Heidelberg, Germany), blue filter cube (excitation filter 405/10, emission filter 460/50), GFP filter cube (excitation filter 470/40, emission filter 525/50) (Leica Microsystem, Heidelberg, Germany), mCherry-T/ET filter cube (excitation filter 578/21, emission filter 641/75, Semrock, Rochester, NY, USA). CoolLED pE-300white (CoolLED Ltd., Andover, UK) with illumination intensity of about 0.04–0.47 W/cm^2^ for a 20× lens and 0.18–2.0 W/cm^2^ for a 40× lens was used as a light source. Images were acquired with an Andor Zyla 5.5 CL 10 Tap sCMOS camera (Andor Technology) controlled by the Micromanager software (ver. 1.4.23) [60].

qPCR of Cre-mediated hTERT cutout was conducted with 10 ng DNA in triplicate in a 25 µL reaction with an qPCR SYBR kit (Evrogen, Moscow, Russia) and a Thermal Cycler CFX96 Real-Time System (Bio-Rad, Hercules, CA, USA) was used. Primer and amplicon sequences for hTERT, Azurite, TurboRFP and the promoter region were from Evrogen and Syntol (Moscow, Russia). This proprietary assay resulted in a 300-bp product within the transgene detected with a SYBR-labeled probe. All reactions consisted of 45 cycles using standard conditions (3 m at 95 °C, and 45 cycles of 15 s denaturation at 95 °C and 3 m annealing at 60 °C and extension at 72 °C).

### 4.16. Flow Cytometry

A Bio-Rad S3 Cell Sorter (Bio-Rad, Hercules, CA, USA) with a 488-nm laser was used for live cell flow cytometry. Fluorescence was detected in the green (FL1, 510–540 nm) and red (FL4, 660–690 nm) channels for the cells expressing eGFP (pCAG-Cre:GFP plasmid) or TurboFP635 (TurboFP635-N vector, Evrogen). A FACSAria cell sorter (BD, Franklin Lakes, New Jersey, NJ, USA) was used for Hoechst 33342 (B2261, Sigma-Aldrich, St. Louis, MO, USA) and TurboFP635 fluorescence detection. The analysis was performed with the AmCyan channel for Hoechst 33342 and the KillerRed channel for TurboFP635. 

### 4.17. Senescence-Associated β-Galactosidase Assay

The cells were seeded onto the wells of 6-well plates (Corning, Corning, New York, NY, USA) in such amounts as to achieve similar confluence across wells. The plates were then incubated for two days.

The cultural medium was removed from the wells before fixation and the cells were rinsed with PBS (PanEco, Moscow, Russia). The cells were fixed with 3% buffered formaldehyde (Biovitrum, Saint Petersburg, Russia) for 3–5 min and then washed with PBS three times for 5 min. After that, an X-gal staining solution containing 1 mg/mL X-gal (Merck, Kenilworth, NJ, USA), 5 mM potassium ferrocyanide, 5 mM potassium ferricyanide, 150 mM sodium chloride, 2 mM magnesium chloride in 40 mM citric acid/sodium phosphate buffer (pH = 6.0) was added to the wells and the plates were incubated at 37 °C for 12–16 h. When the incubation was finished, the cells were washed with PBS three times for 5 min.

Microscopic images of the cells were then obtained using a bright-field light microscope EVOS FL AUTO under the same conditions for all images. For analyzing staining intensity, the images were subjected to color deconvolution using a FiJi plugin [61]. X-gal staining was separated from the Region Of Interest (ROI) (R: 0.73737514, G: 0.4551425, B: 0.49912247). The areas covered with the cells were selected on the original images using the PHANTAST FiJi plugin [56]. Staining intensity was then estimated as the mean pixel intensity of the image with color deconvolution on the area covered with the cells.

### 4.18. Cell Proliferation

ICC analysis with anti-Ki-67 antibodies was used for estimation of fibroblast proliferation. Hoechst 33342 was used for nuclei detection. The staining protocol is described above. Fluorescent images were made using an EVOS FL AUTO microscope. Nuclei counting was performed using the FiJi software [54].

The estimation of keratinocyte proliferation was made using Click-iT™ Plus EdU Flow Cytometry Assay Kit (Invitrogen, Carlsbad, CA, USA) according to the manufacturer’s instructions. The keratinocytes were labeled with EdU at the concentration of 10 μM for 20 h and analyzed using a Bio-Rad S3 Cell Sorter (Bio-Rad, Hercules, CA, USA).

### 4.19. Secretome Analysis

#### 4.19.1. Preparation of a Sample with Total Secreted Proteins for Mass Spectrometric Analysis

For each line, 100 thousand keratinocytes were passed in a 35-mm Petri dish and cultivated for four days in the CnT-07 medium (CELLnTEC, Bern, Switzerland). The conditioned medium was collected in tubes and centrifuged at 12,000 rcf for 10 min to precipitate the cellular debris. The supernatant was transferred to new tubes. Protein concentration was measured using the Bradford method with the Coomassie G-250 solution (Bio-Rad, Hercules, CA, USA) with the help of a Bio-Rad iMark Microplate Reader. The conditioned medium was concentrated on Amicon Ultra Spin Columns 10k (Merck, Kenilworth, NJ, USA) according to the manufacturer’s protocol. After concentrating the complete medium volume, 100 mM Tris-HCl (pH = 8) was applied to the column twice.

#### 4.19.2. Preparation of Samples for Secretome Analysis

Reduction, alkylation and digestion of the proteins were performed as described in [62] with minor modifications. Briefly, 20 μL of the sodium deoxycholate (SDC) reduction and alkylation buffer, pH 8.5, containing 100 mM Tris, 1% (*w*/*v*) SDC, 10 mM Tris(2-Carboxyethyl)Phosphine TCEP and 20 mM 2-chloroacetamide were added to a 20 g protein sample. The sample was sonicated in an ultrasonic water bath for 1 min, heated at 95 °C for 10 min, cooled to room temperature and the equal volume of trypsin solution in 100 mM Tris, pH 8.5, was added in a 1:50 (*w*/*w*) ratio. After overnight digestion at 37 °C, peptides were acidified with 40 μL of 2% trifluoroacetic acid (TFA) mixed with 80 μL of ethyl acetate and loaded on polystyrene-divinylbenzene, reversed-phase sulfonate (SDB-RPS) StageTips containing two 14-gauge SDB-RPS plugs, and the StageTips were centrifuged at 300 g until all of the solution would go through the StageTips (typically 5 min). After washing the StageTips with 100 μL of a 1% TFA/ethyl acetate 1:1 mixture (two times) and 100 μL of 0.2% TFA, the peptides were eluted in a clean tube with 50 μL of a 50% acetonitrile/5% ammonia mixture using centrifugation at 300 g. The collected material was vacuum-dried and stored at −80 °C. Before the analyses, the peptides were dissolved in a 2% acetonitrile/0.1% TFA buffer at a concentration of 0.5 μg/μL and sonicated for 1 min.

#### 4.19.3. Liquid Chromatography and Mass Spectrometry

The samples were loaded to a home-made trap column, 20 × 0.1 mm, packed with the Inertsil ODS3 3 m sorbent (GLSciences, Tokyo, Japan) in the loading buffer (2% ACN, 98% H_2_O, 0.1% TFA) at 10 μL/min flow and separated at room temperature in a home-packed [63] fused-silica column, 300 × 0.1 mm, packed with Reprosil PUR C18AQ 1.9 (Dr. Maisch) into an emitter prepared with a P2000 Laser Puller (Sutter, Atlanta, GA, USA). Reverse-phase chromatography was performed with an Ultimate 3000 Nano LC System (ThermoFisher Scientific, Waltham, MA, USA), which was coupled to an Orbitrap Lumos Tribrid mass spectrometer (ThermoFisher Scientific, Waltham, MA, USA) via a nanoelectrospray source (ThermoFisher Scientific, Waltham, MA, USA). The peptides were loaded in a loading solution (98% 0.1% (*v*/*v*) formic acid, 2% (*v*/*v*) acetonitrile) and eluted with a linear gradient: 3–6% solution B (0.1% (*v*/*v*) formic acid, 80% (*v*/*v*) acetonitrile) for 3 min; 6–25% solution B for 30 min, 25–40% solution B for 25 min, 40–60% solution B for 4 min, 60% solution B for 3 min, 60–99% solution B for 0.1 min, 99% solution B during 10 min, 99–3% solution B for 0.1 min at a flow rate of 500 nL/min. After each gradient, the column was re-equilibrated with solution A (0.1% (*v*/*v*) formic acid, 2% (*v*/*v*) acetonitrile) for 10 min. MS data were collected in the Data-Dependent Analysis (DDA) mode. MS1 parameters were as follows: 60 K resolution; 350–1600 scan range; max injection time, Auto; Automatic Gain Control (AGC) target, standard. Ions were isolated with a 1.2 *m*/*z* window, preferred peptide match and isotope exclusion. Dynamic exclusion was set to 30 s. MS2 fragmentation was carried out in the High-energy Collisional Dissociation (HCD) mode at 7.5 K resolution with HCD collision energy of 30%, automatic maximum injection time, AGC target—standard. Other settings: charge exclusion—unassigned, 1, > 7.

#### 4.19.4. Secretome Data Analysis

Raw spectra were processed using MaxQuant 1.6.6.0 (MQ) [64] and Perseus [65]. The data were searched against the Human Uniprot Tremble database containing canonical and isoform proteins, version from September 2020.

MaxQuant search was performed with the default parameter set, including Trypsin/p protease specificity, maximum two missed cleavages, Met oxidation, protein N-term acetylation and Asn/Gln (NQ) deamidation as variable modifications and carbamidomethyl Cys as a fixed modification, maximum five modifications per peptide, 1% peptide-spectra match (PSM) and protein false discovery rate (FDR). The following options were turned on: second peptide, max label-free quantification (maxLFQ), the match between runs. All the runs were analyzed as independent experiments and processed in Perseus.

In Perseus, the protein group results were filtered for contaminants, reverse and “identified only by site” proteins. Only the proteins with maxLFQ values in at least three out of seven LC–MS runs were used. For them, the missing values were imputed from a normal distribution with 0.3 intensity distribution sigma width and 1.8 intensity distribution center downshift. The two-sample *t*-test with permutation-based FDR (5%) was applied to search for significantly changing proteins. The GO analysis was performed using STRING [24] and Phantasus v1.9.2 [25] according to the guide [66].

### 4.20. Software

Statistical calculations were made using Microsoft Excel 2016 (Microsoft, Redmond, Washington, WA, USA), Origin 8 (OriginLab Corporation, Northampton Massachusetts, MA, USA) and GraphPad Prism 8 (GraphPad Software, San Diego, CA, USA).

Image preparation and analysis were performed using PaintNET 4.2.15 (Microsoft, Redmond, Washington, WA, USA), Adobe Photoshop CS6 13.0.1.3 (Adobe, San Jose, California, CA, USA), InkScape 1.0.2 and FiJi, GM 1.2 (SoftGenetics, LLC, State College, PA, USA).

Flow cytometry files were analyzed using FlowJo X 10.0.7 (BD, Franklin Lakes, New Jersey, NJ, USA).

### 4.21. Statistical Analysis

In the TRAPeze assay, 95% confidence intervals were used to compare cell lines. Morphological parameters in microscopy data were compared using the two-sample *t*-test with the 0.05 significance level. For comparison of flow cytometry data, the FlowJo Chi Squared comparison or one-way ANOVA with the Tukey’s test with the 0.05 significance level were used. For comparison of collagen gel contraction, the Mann–Whitney test with the 0.05 significance level was used. Data of the SA-β-gal assay were compared by the Mann–Whitney test or the two-sample *t*-test with the 0.05 significance level and 95% confidence intervals were calculated as well.

## 5. Conclusions

Here, the establishment of new RDEB-specific stable cell lines was reported. High activity of hTERT in these lines ensured the extended lifetime for the panel of fibroblast lines. The new method of combined hTERT/BMI-1 ectopic expression was applied for immortalization of epidermal keratinocyte cell lines. Cre-mediated site-directed recombination leads the cells to get rid of the transgenic hTERT and turns themselves closer to their primary status. Morphology and functional tests of immortalized cell lines confirmed the phenotypes similar to those of the primary cells for both the fibroblast and keratinocyte panels. New immortalized keratinocyte lines, both normal and RDEB ones, were able to form in vitro multilayered stratified epidermal equivalents with adequate distribution of differentiation markers. The unbiased investigation of immKEB- and immKHC-secreted proteins revealed the pattern of differences in the ECM-associated proteins, dermal-epidermal junction components, the inhibition of several proteases and enhanced expression of protease inhibitors. Several tumor-supporting markers were found to downregulate in both immKEB lines that pointed out the anticancer traits of RDEB keratinocytes at this state of disease progression. We conclude that new immortalized panels of cell lines are useful for substitution of primary cell lines, thereby representing a biologically relevant model for in vitro investigation of processes involved in RDEB development.

## Figures and Tables

**Figure 1 ijms-22-03809-f001:**
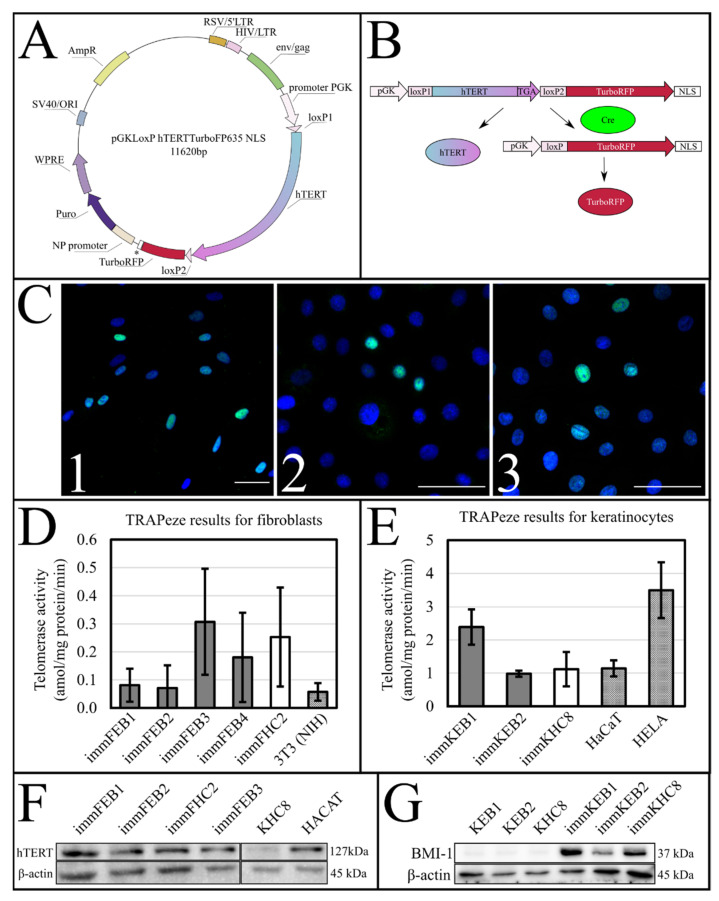
(**A**) The scheme of lentiviral plasmid pGKLoxP hTERT TurboFP635 NLS. * NLS—nuclear localization signal of simian virus 40 (SV40). Complementary DNA of human telomerase (hTERT) with the stop codon is floxed by LoxP (locus of X-over P1), fluorescent protein TurboFP635NLS is not translated unless Cre is cut out of hTERT. LTR—long terminal repeats. ORI—origin of replication. WPRE—woodchuck hepatitis post-transcriptional regulatory element. NP—promoter of Homo sapiens tumor protein p53. (**B**) The scheme of Cre-mediated site-directed recombination of the LoxP hTERT TurboFP635 NLS template. (**C**) Immunocytochemistry (ICC) analysis of immortalization; nuclei stained with DAPI (blue); confocal microscopy; scale bar—50 μm; 1—immFEB1 staining against hTERT (green); 2—immKHC8 staining against hTERT (green); 3—immKHC8 staining against BMI-1 (green). (**D**) Quantitation of telomerase activity in primary and immortalized fibroblast cell lines. Telomerase activity in primary cell lines not shown because it is equal to zero. The mean value of telomerase activity and 95% confidence level are given for each cell line. Cell line 3T3 (NIH) is the positive control. (**E**) Quantitation of telomerase activity in primary and immortalized keratinocyte cell lines. Telomerase activity in primary cell lines not shown because it is equal to zero. The mean value of telomerase activity and 95% confidence level are given for each cell line. HaCaT and HELA cell lines are the positive control. (**F**) Western blot analysis of hTERT expression. KHC—healthy keratinocyte control. HaCaT cells were taken as the positive control. Electrochemiluminescence (ECL) detection, 10% SDS-PAGE, anti-hTERT monoclonal antibody (upper panel), anti-ꞵ-actin antibody polyclonal antibody (lower panel). Band weights for hTERT and β-actin fall within the range 127 kDA ± 5% and 45 kDA ± 5%, respectively. For uncut and unadjusted images, see Appendix A. (**G**) Western blot analysis of BMI-I expression. ECL detection, 10% SDS-PAGE, anti-BMI-I monoclonal antibody (upper panel), anti-β-actin antibody polyclonal antibody (lower panel). For uncut and unadjusted images, see Appendix A.

**Figure 2 ijms-22-03809-f002:**
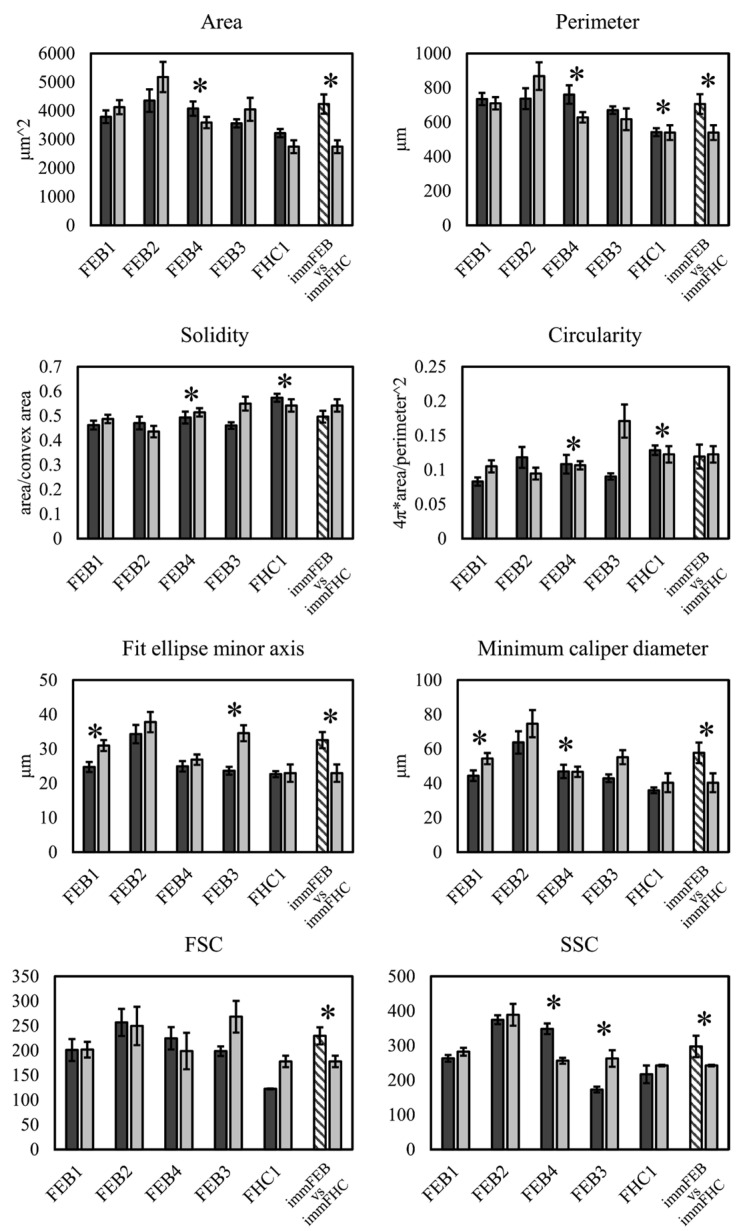
Morphological analysis of phase contrast microscopy (area, perimeter, solidity, circularity, fit ellipse minor axis, minimum caliper diameter) and flow cytometry forward scatter (FSC) and side scatter (SSC) data of primary (left dark grey column) and immortalized (right light grey column) fibroblasts. Mean value and standard error are shown for each cell line. Shaded columns are combined data for four immortalized RDEB fibroblasts represented as mean of the means with standard error. Asterisks (*) indicate significant differences between left and right columns (*p* < 0.05). Morphological parameters were compared by the two-sample *t*-test. Flow cytometry data were compared by the FlowJo Chi Squared comparison: FSC (T(x) = 342, baseline T(x) = 115); SSC (T(x) = 1326, baseline T(x) = 43) (immFEB vs. immFHC) and one-way ANOVA with the Tukey’s test (primary vs. immortalized).

**Figure 3 ijms-22-03809-f003:**
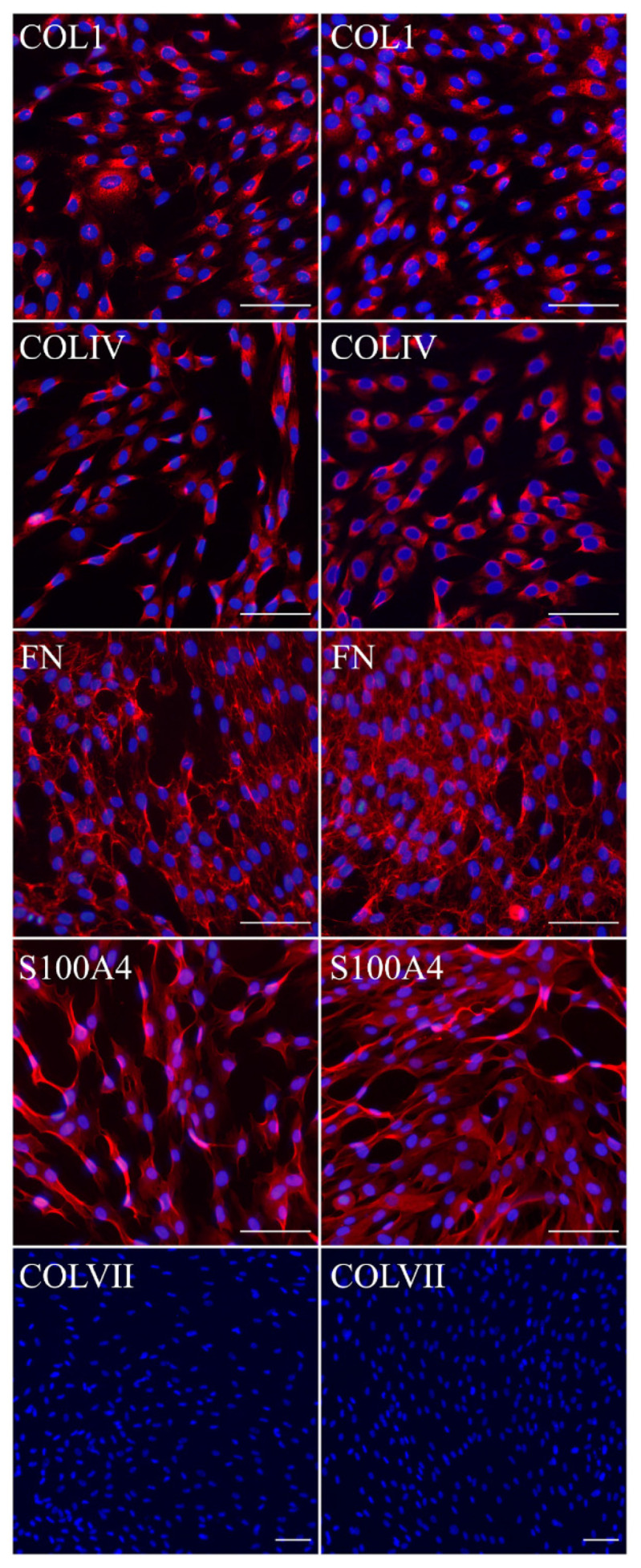
Immunocytochemistry of FEB1 (left column) and immFEB1 (right column). Red channel—markers: COLI—collagen type I; COLIV—collagen type IV; FN—fibronectin; S100A4—fibroblast-specific protein 1; COLVII—collagen type VII. Blue channel—nuclei (DAPI). Scale bar—100 µm. Confocal microscopy (COLI, COLIV, FN, S100A4) and fluorescence microscopy (COLVII).

**Figure 4 ijms-22-03809-f004:**
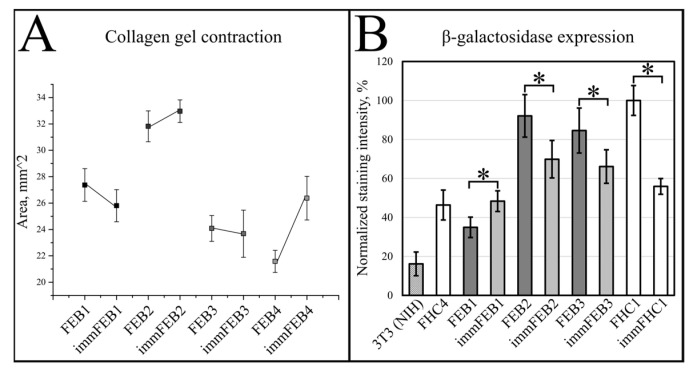
(**A**) The results of collagen gel contraction assay for FEB and immFEB. The smaller the area of the gel, the higher the cell contraction recorded. Means plot. Error—standard error. (**B**) Senescence-Associated-β-galactosidase (SA-β-gal) staining in fibroblasts. The most intensely stained cells were taken as 100%. Asterisks (*) indicate a significant difference (*p* < 0.05) between primary and immortalized cell lines when compared by the Mann–Whitney test. The mean value of normalized staining intensity and 95% confidence level are given for each cell line. Positive control—3T3 (NIH). An example of staining can be found in Appendix A.

**Figure 5 ijms-22-03809-f005:**
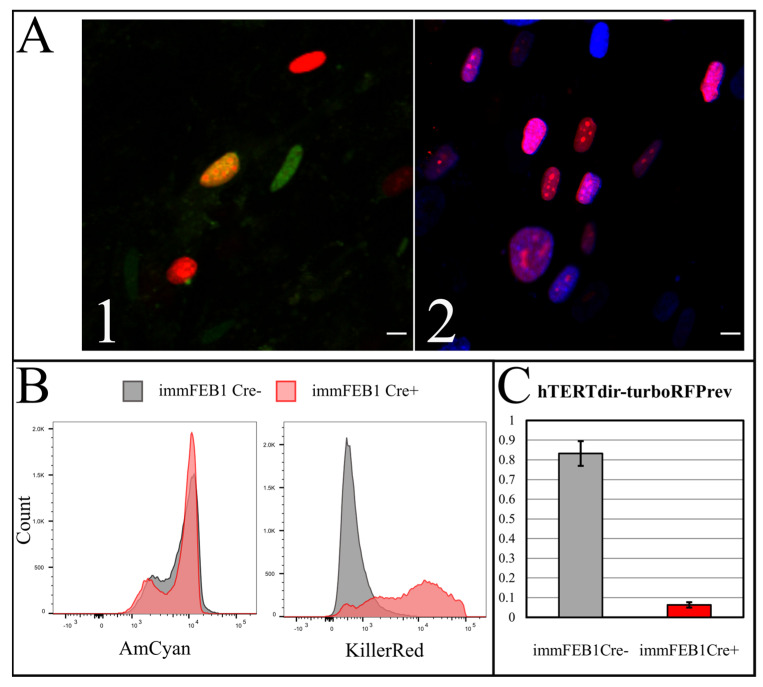
(**A1**) ImmFEB1 with stable expression of hTERT–TurboFP635 after transfection with pCre-eGFP; green channel—Cre-eGFP, red channel—TurboFP635NLS. Live cell labeling. Confocal microscopy. Scale bar—10 µm. (**A2**) ImmFEB1 transfected with pCre-eGFP after fluorescence-activated cell sorting (FACS) for TurboFP635-positive cells; blue channel—DNA (DAPI), red channel—TurboFP635. Immunofluorescence imaging. Confocal microscopy. Scale bar—10 µm. (**B**) FACS analysis of immFEB1. Left chart: AmCyan excitation revealed fluorescence of nuclei stained with Hoechst 33342; right chart: KillerRed excitation revealed fluorescence of TurboFP635 before (grey color, Cre-) and after (pink color, Cre+) transfection with Cre-eGFP. (**C**) Quantitative PCR analysis of the hTERT presence of immFEB1 before and after Cre-mediated recombination. The ordinate value is the relative amount of hTERT–TurboFP635 junction amplicons measured in reference to the amount of respective TurboFP635 amplicons. The mean value and standard deviation are given for each cell line.

**Figure 6 ijms-22-03809-f006:**
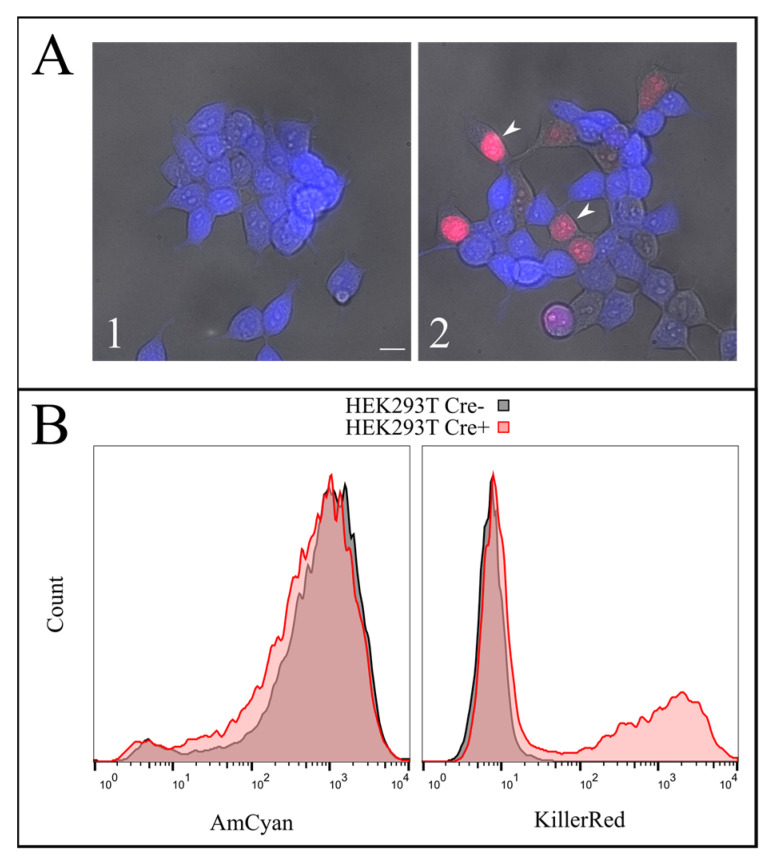
(**A1**) Human Embryonic Kidney 293T cell line (HEK293T) with stable expression of Azurite. The Azurite signal has only cytoplasmic localization. (**A2**) HEK293T with the expression of TurboFP635NLS one week after transfection with pCre-eGFP. Blue channel—Azurite, red channel—TurboFP635. Arrows indicate TurboFP635+ cells with nuclear localization. Live cell labeling. Fluorescent microscopy. Scale bar—10 µm. (**B**) FACS analysis of HEK293T with stable expression of Azurite and with the expression of TurboFP635NLS one week after transfection with pCre-eGFP. Left chart: AmCyan excitation revealed fluorescence of Azurite; right chart: Killer Red excitation revealed fluorescence of TurboFP635; before (grey, Cre−) and after (red, Cre+) transfection with Cre-eGFP.

**Figure 7 ijms-22-03809-f007:**
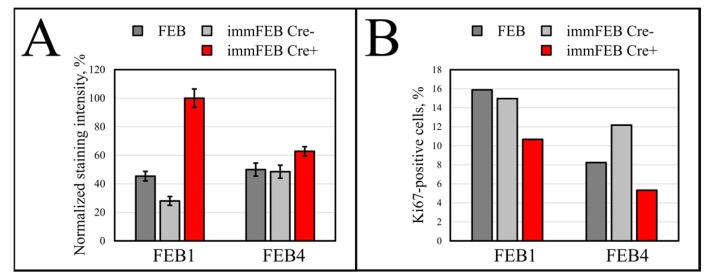
(**A**) Senescence-Associated beta galactosidase (SA-β-gal) normalized staining intensity of FEB and immFEB before (Cre−) and after (Cre+) immortalization and transfection with Cre-eGFP. The most intensely stained cells were taken as 100%. The difference between cell lines was significant (*p* < 0.05) when compared by the two-sample *t*-test (with the exception of FEB4 and immFEB4 Cre−). The mean value of SA-β-gal-normalized staining intensity and 95% confidence level are given for each cell line. (**B**) Proliferation of FEB and immFEB before (Cre−) and after (Cre+) immortalization and transfection with Cre-eGFP.

**Figure 8 ijms-22-03809-f008:**
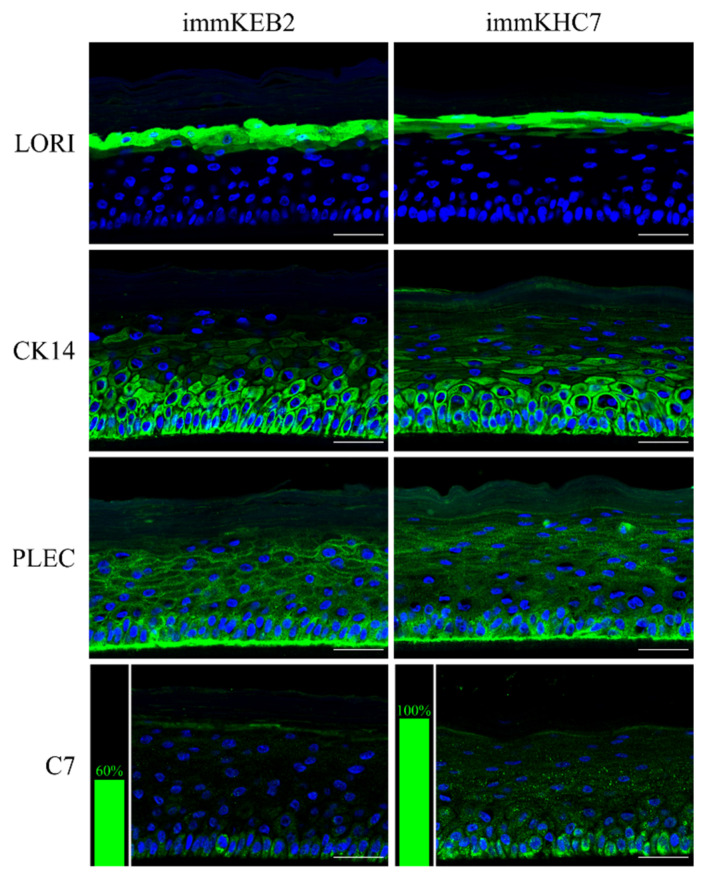
Immunohistochemistry (IHC) of reconstructed epidermis from immKEB2 (RDEB Inversa) and immKHC. Nuclei stained with DAPI (blue). Confocal microscopy. Scale bar—50 μm. Markers (green): LORI—loricrin, CK14—cytokeratin 14, PLEC—plectin, C7—type VII collagen. Histogram shows staining intensity of C7 in the basal layer of immKEB relative to immKHC.

**Figure 9 ijms-22-03809-f009:**
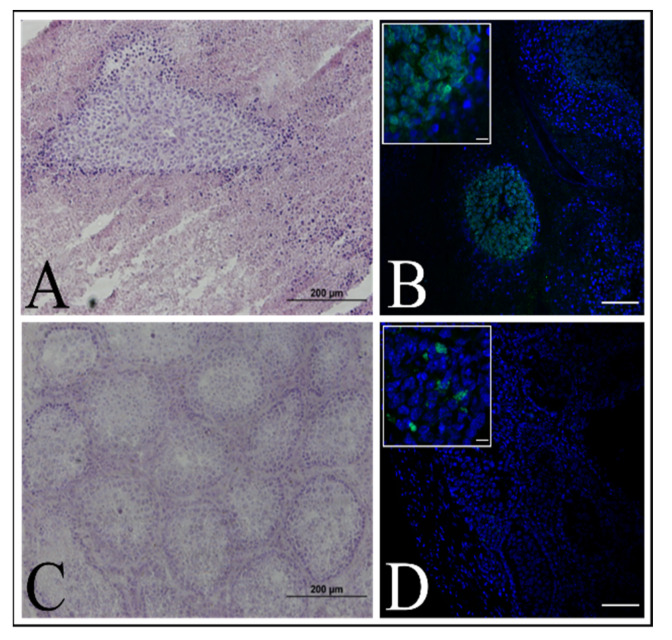
Tumor formation test by cell transplantation into nude mouse testes. Cryosection thickness is 9 μm. (**A**,**C**) Hematoxylin–eosin staining. Bright-field microscopy. Scale bar—200 μm. (**B**,**D**) Immunohistochemistry. Nuclei (DNA) stained with DAPI (blue). Human nuclei stained with antibodies (green). Confocal microscopy. Scale bar—100 μm (main image), 10 μm (insert). (**A**,**B**) Control mouse testis tissue with a tumor four weeks after HELA transplantation. (**C**,**D**) Mouse testis tissue with immKEB2 cells ten weeks after transplantation. Normal testis morphology. The insert showed the remaining immKEB2 cells immunostained with human nuclear antigen.

**Figure 10 ijms-22-03809-f010:**
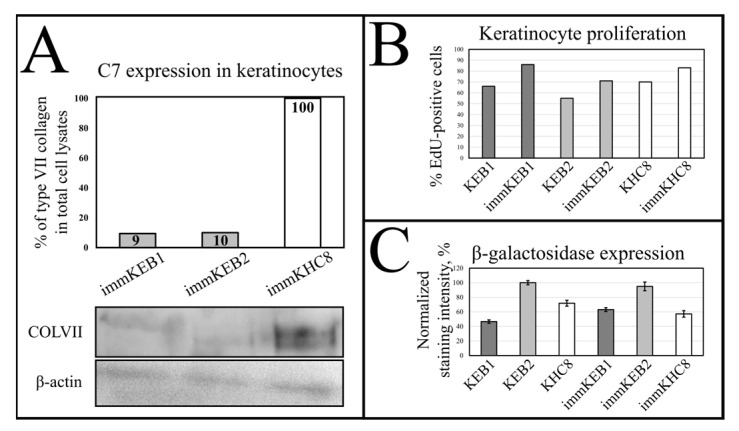
(**A**) Western blot analysis of total cell lysates of immortalized keratinocytes; anti-type VII collagen polyclonal antibody (upper panel), anti-β-actin (lower panel). For uncut and unadjusted images, see Appendix A. (**B**) Keratinocyte proliferation measured by 5-Ethynyl-2-Deoxyuridine (EdU) labeling and flow cytometry. (**C**) SA-β-gal-normalized staining intensity of keratinocytes. The most intensely stained cells were taken as 100%. The difference between cell lines was significant (*p* < 0.05) when compared by the Mann–Whitney test (with exception of KEB2 and immKEB2). The mean value of SA-β-gal normalized staining intensity and 95% confidence level are given for each cell line. An example of staining can be found in Appendix A.

**Figure 11 ijms-22-03809-f011:**
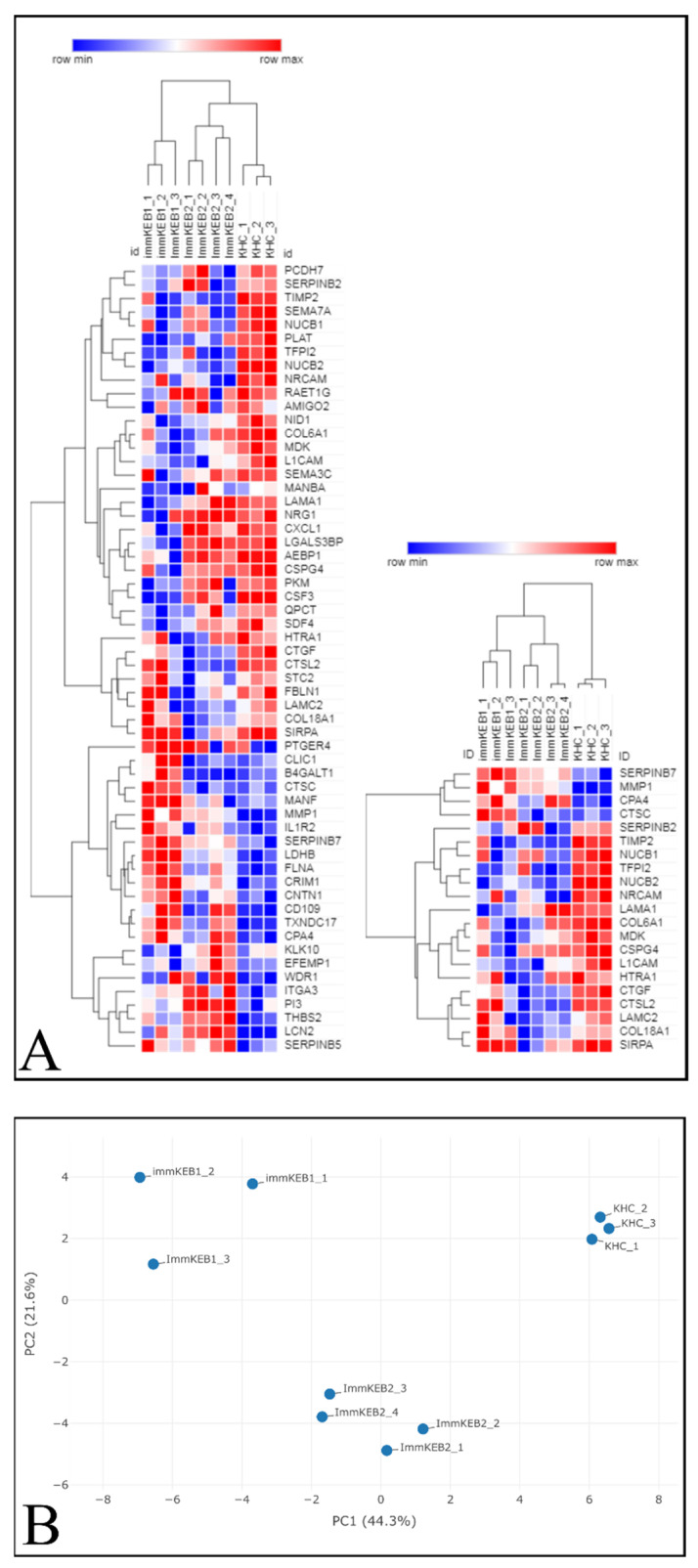
(**A**) Heatmaps of differentially expressed proteins (DEPs) (immKEB1, immKEB2, immKHC). Left panel: DEPs in either immKEB1 or immKEB2 lines relative to immKHC, immKEB1 + immKEB2_immKHC, named 1 + 2_3. DEPs of immKEB1 or immKEB2 in comparison with immKHC. Biological repeats are denoted in columns above as KEB1_1, KEB1_2, KEB1_3. Right panel: DEPs specific to both immKEB lines relative to immKHC, immKEB1 ∩ immKEB2_immKHC, named 1 ∩ 2_3. Biological repeats are denoted as KEB1_1, KEB1_2, KEB1_3. (**B**) Principal component analyses (PCA) of 1 + 2_3 data. Each point denotes one replicate.

**Table 1 ijms-22-03809-t001:** List of patients with the recessive form of dystrophic epidermolysis bullosa (RDEB) and healthy control donors, gender and age information, numbers of patient-specific cell lines obtained.

Patient	Gender/Age	Cell Lines
d1 RDEB	M/8	FEB1, immFEB1, KEB1, immKEB1
d2 RDEB	M/16	FEB2, immFEB2 KEB2, immKEB2
d3 RDEB	M/5	FEB3, immFEB3
d4 RDEB	M/21	FEB4, immFEB4
d5	F/58	FHC1, immFHC1
d6	F/57	FHC2, immFHC2
d8	F/24	FHC4
d12	M/8	KHC8, immKHC8

FEB and immFEB—primary and immortalized fibroblasts from RDEB patients, FHC and immFHC—primary and immortalized fibroblasts from healthy donors, KEB and immKEB—primary and immortalized keratinocytes from RDEB patients, KHC and immKHC—primary and immortalized keratinocytes from healthy donors.

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
