# Peer review of "hTERT-Driven Immortalization of RDEB Fibroblast and Keratinocyte Cell Lines Followed by Cre-Mediated Transgene Elimination"

_ijms, 2021, doi:10.3390/ijms22083809_

Round 1

Reviewer 1 Report

I found the paper interesting and well written. I have no suggestion for improvement. The data provided in the paper seem valid. The immortalization of keratinocytes and fibroblasts derived from patients with dystrophic epidermolysis bullosa (RDEB) that display the properties of the original source, that is not altered by the immortalization process, is useful for studying the effect of the individual genetic alterations that occur in patients with this disease. Immortalized cells are much easier to work with and this is especially true for keratinocytes that have limited in vitro lifespan. The secretome analysis that they provide convinced me that indeed their approach results in useful findings toward understanding the various forms of this disease. Constructing 3D skin models with these cells will also give a useful in vitro model to find ways by which the lack of collagen VII could be influenced. I am aware that any in vitro manipulation of cells can not directly transfer to the in vivo situation, however with proper controls it is possible to check the in vivo relevance of any in vitro result. In short, I found the article publishable as it is. 

Author Response

We would like  to thank Reviewer 1 for  the valuable estimation of our work.  

Reviewer 2 Report

This manuscript reports immortalization skin cells. Fibroblasts and keratinocytes from patients with the recessive form of dystrophic epidermolysis bullosa (RDEB) immortalized with h-TERT followed by Cre-mediated elimination of the transgene. The authors used several approaches to characterize the cell lines. This study is not a major scientific breakthrough but it is interesting for researchers in this field.

The text provides a lot of qualitative interpretation but little quantification of results. The writing should be improved, especially abstract and results. The abstract should describe more clearly which experiments were done in this study and what were the results. The main text is overly long.

Results of statistical analysis are not reported appropriately in this manuscript. Figures show bars and error bars, but it is not described if there are “real” differences. In the Materials and Methods section, “Statistical calculations” are inappropriately reported as part of the paragraph “software”. The statistical analysis must be substantially improved.

Cre-mediated recombination should be confirmed by PCR and sequencing.

Did the proteomic analysis of the secretomes show reduced expression of collagen 7 in cell lines carrying the mutation?

Author Response

“The text provides a lot of qualitative interpretation but little quantification of results. The writing should be improved, especially abstract and results. The abstract should describe more clearly which experiments were done in this study and what were the results. 

The new version of abstract contains information about the majority of the experiments in the article and their results.

We made the new version of abstract

“The main text is overly long.”

We made the changes in the main text, most of them affected the section 3 of the Discussion.

“Results of statistical analysis are not reported appropriately in this manuscript. Figures show bars and error bars, but it is not described if there are “real” differences. In the Materials and Methods section, “Statistical calculations” are inappropriately reported as part of the paragraph “software”. The statistical analysis must be substantially improved.”

All changes were highlighted in the main text.

We changed the section “2.3.1. General Morphology”. It now includes information about used statistical tests; significance is shown on the Figure 2. The significance is also shown on the Figure 4B. The new “Statistical analysis” section is added in Materials and Methods.

General Morphology of fibroblasts

We have investigated the impact of immortalization on morphology of the fibroblasts. From 43 to 127 cells from each cell line were measured for morphological analysis. Morphology parameters of primary and immortalized cells from the same donors were compared by the two-sample t test. Forward scatter (FSC) and side scatter (SSC) were measured in triplicates for each cell line and were compared by one way ANOVA with Tukey test.

To investigate if immFEB differ from immFHC we compared combined data from four immFEB to immFHC by the two-sample t test for microscopy data and by the FlowJo Chi Squared comparison for flow cytometry data.

Despite that, some cell lines show differences in some morphological parameters there is no clear dependence in any parameter between morphology and immortalization. For the most we can say that immortalization does not have a significant impact on fibroblasts morphology therefore maintaining differences between FEB and FHC like increased area, perimeter, minimum fit ellipse axis, minimum caliper diameter, size and granularity for FEB (Figure 2).

Senescence associated-β-galactosidase staining in fibroblasts

Figure 4. (A) The results of collagen gel contraction assay for FEB and immFEB. The smaller the area of the gel, the higher the cell contraction recorded. Means plot. Error - standard error. The difference between primary and immortalized cell lines is not significant (p>0.05) when compared by Mann-Whitney test (B) Senescence associated-β-galactosidase (SA-β-gal) staining in fibroblasts. The most intensely stained cells were taken as 100%. Asterisks indicate significant difference (p<0.05) between primary and immortalized cell lines when compared by Mann-Whitney test. The mean value of normalized staining intensity and 95% confidence level are given for each cell line. 3T3 (NIH) is positive control. An example of the staining can be found in Supplementary figure S4.

Statistical analysis

In TRAPeze assay 95% confidence intervals were used to compare cell lines.

Morphological parameters in microscopy data were compared using two-sample t-test with 0.05 significance level.

For comparison of flow cytometry data (FSC and SSC) the FlowJo Chi Squared comparison or one way ANOVA with Tukey test with 0.05 significance level were used.

For comparison of collagen gel contraction Mann-Whitney test with 0.05 significance level was used.

Data of SA-β-gal assay were compared by Mann-Whitney test or two-sample t-test with 0.05 significance level. 95% confidence intervals were calculated as well.

Did the proteomic analysis of the secretomes show reduced expression of collagen 7 in cell lines carrying the mutation?

Collagen type VII alpha 1 chain is non-significantly diminished in immKEBs compared to immKHC. The data about Col7A1 could be found on the Supplementary Table S2, (page 1, 3, line 215, LogFC=0,1), the differences between the groups in Col7A1 content are not statistically significant according to the paired t-test (two-sample t-test based FDR of 5% was applied, as described in Section 4.19.4.  Secretomes Data analysis). However, mass spectrometric data do not reveal the amount of functional procollagen VII homotrimers and consequently of the anchoring fibrils. The pathogenesis of the disease is determined by the absence of a functional C7 protein, causing a dramatic misalignment of the expression profile in fibroblasts and keratinocytes, leading to chronic inflammation and transformation. 

Cre-mediated recombination should be confirmed by PCR and sequencing.

We performed the amplification from genome DNA after Cre-mediated recombination from both immFEB1 Cre+ and HEK293T Azuritere Cre+ cells. We revealed the full–length form of the transgenes, which was about 6 kb from pGK promoter to Puromicin gene. At the same time, we observed the suppression of PCR amplification of pgK – TurboFP635transgene possibly due to a presence of the hairpin structure. Nevertheless, after testing four different specific primer pairs we selected the best one and performed additional amplification with nested specific primers. This allows to obtain the specific bands with 1000 and 700 base pairs, respectively (Supplementary Figure S5). Sequencing of these products confirmed the presence of TurboFP635 (see below, Fig A), pGK promoter (see below, Fig B) and the middle part of the TurboFP635 sequence (see below, Fig C). The last chromatogram revealed the double peaks which illustrate the hurdled place of the sequence due to the possible hairpin structure, which hampers PCR amplification.  

Figure S5. PCR amplification of DNA from HEK293T Azurite cells  after Cre-mediated recombination. Line 1- DNA ladder. Line 2 - PCR with pGKdir5, Turborev end primers; line 3 - PCR with nested primers pGKdir 4, TurboFP635rev2. The first amplification with pGKdir5 TurboFP635rev end primers produced the band with the expected length of about 1 kb. The product shown in line 3 was obtained from the first product with nested specific primers Turbo-rev2 and pGKdir4, revealing the band of 700 base pairs, expected to result from cutting out the target construct. Line4 -DNA ladder.

Then we attached:

  1. Sequence of TurboFP635
  2. Sequence of pGK promoter
  3. The end of pGK promoter; the start of TurboFP635; The middle part of the TurboFP635 sequence.

Next, we performed the real-time Q-PCR of genomic DNA isolated from immFEB1 cells before and after Cre-eGFP transfection to estimate the amount of hTERT transgene (Figure 5C). We used primer pairs for the junction of hTERT- TurboFP635 and specific only to TurboFP635 to discriminate between these forms in the genome.

Finally, we are very grateful to the Reviewer for the valuable comments, which permit us to improve our manuscript.

Round 2

Reviewer 2 Report

The manuscript was improved.